# Zero-shot Transfer Learning within a Heterogeneous Graph via Knowledge Transfer Networks

**Minji Yoon**[*]
Carnegie Mellon University

**John Palowitch**
Google Research

**Dustin Zelle**
Google Research

**Ziniu Hu**[*]
University of California Los Angeles

**Ruslan Salakhutdinov**
Carnegie Mellon University

**Bryan Perozzi**
Google Research

## Abstract

Data continuously emitted from industrial ecosystems such as social or e-commerce platforms are commonly represented as heterogeneous graphs (HG) composed of multiple node/edge types. State-of-the-art graph learning methods for HGs known as heterogeneous graph neural networks (HGNNs) are applied to learn deep context-informed node representations. However, many HG datasets from industrial applications suffer from label imbalance between node types. As there is no direct way to learn using labels rooted at different node types, HGNNs have been applied on only a few node types with abundant labels. We propose a zero-shot transfer learning module for HGNNs called a Knowledge Transfer Network (KTN) that transfers knowledge from *label-abundant* node types to *zero-labeled* node types through rich relational information given in the HG. KTN is derived from the theoretical relationship, which we introduce in this work, between distinct *feature extractors* for each node types given in a HGNN model. KTN improves performance of 6 different types of HGNN models by up to $960\%$ for inference on zero-labeled node types and outperforms state-of-the-art transfer learning baselines by up to $73\%$ across 18 different transfer learning tasks on HGs.

## 1 Introduction

Large technology companies commonly maintain large relational datasets, derived from their internal logs, that can be represented as or joined into a massive heterogeneous graph (HG) composed of nodes and edges with multiple types (30). For instance, in e-commerce networks, there are product, user, and review nodes, all interconnected by many edge types that represent forms of interactions such as spending (user-product), reviewing (user-review), and reviews-of (product-review). To learn powerful features representing the complex multimodal structure of HGs, various heterogeneous graph neural networks (HGNN) have been proposed (15; 26; 35; 43).

A common issue in these industrial applications of HGNNs is the label imbalance among different node types. For instance, publicly available *content* nodes – such as those representing video, text, and image content – are abundantly labelled, whereas labels for other types (such as *user* or *account* nodes) may be much more expensive to collect (or even not available, e.g. due to privacy restrictions). This means that in most standard training settings, HGNN models can only learn to make good inferences for a few label-abundant node types, and can usually not make any inferences for the remaining node types, given the absence of any labels for them.

If there is a pair of *label-abundant* and *zero-labeled* node types which share an inference task, could we transfer knowledge between them? One body of work has focused on transferring knowledge between

---

[*]Work done while interning at Google

36th Conference on Neural Information Processing Systems (NeurIPS 2022).

nodes of the *same* type from two *different* HGs (i.e., graph-to-graph transfer learning) (16; 40). However, these approaches are not applicable in many real-world scenarios for three reasons. First, any external large-scale HG that could be used in a graph-to-graph transfer learning setting would almost surely be proprietary. Second, even if practitioners could obtain access to an external industrial HG, it is unlikely the distribution of that (source) graph would match their target graph well enough to apply transfer learning. Finally, node types suffering label scarcity are likely to suffer the same issue on other HGs (e.g. user nodes).

In this paper, we introduce a zero-shot transfer learning approach for a *single* HG (assumed to be fully-owned by the practitioners), transferring knowledge from labelled to unlabelled node types. This setting is distinct from any graph-to-graph transfer learning scenarios, since the source and target domains exist in the same HG dataset, and are assumed to have different node types. Our model utilizes the shared context between source and target node types; for instance, in the e-commerce network, the latent (unknown) labels of user nodes can be strongly correlated with spending/reviewing patterns that are encoded in the cross-edges between user nodes and product/review nodes. We propose a novel zero-shot transfer learning problem for this HG learning setting as follows:

**Informal Problem Definition 1. Zero-shot cross-type transfer learning running on a HG:**
*Given a heterogeneous graph $\mathcal{G}$ with node types $\{s, t, \cdots\}$ with abundant labels for source type $s$ but no labels for target type $t$, can we train HGNNs to infer the labels of target-type nodes?*

A naïve solution to this problem would be to re-use an HGNN pre-trained on the source nodes for target node inference, given that both domains exist in the same HG. However, as we show in our paper, HGNNs have distinct parameter sets for each node type (15), edge type (26), and meta-path type (8; 35). These facts cause HGNNs to learn entirely different *feature extractors* for nodes and edges of different types – in other words, the final embeddings for source and target nodes are computed by different sets of parameters in HGNNs. Thus, a classifier pre-trained on source nodes will fail to perform well on inference tasks for target nodes. The field of domain adaptation (DA) targets this setting, seeking to transfer knowledge from a source domain with abundant labels to a target domain which lacks them (9; 19; 20; 27). However, distinct feature extractors across node types in HGNNs break a standard assumption of DA setting, namely that source and target domains share the same feature extractors (e.g., CNNs for both source and target image domains). As we demonstrate in this paper, in our problem setting, DA approaches fail to achieve the outstanding performance they are known for in computer vision and NLP.

In our work, we first dissect the gradient path of HGNN models to see how feature extractors are designed independently for each node type, and some empirical consequences. Then we theoretically analyze how feature extractors across node types relate to each other and how their output distributions could be represented in terms of each other. We model this theoretical relationship between two feature extractors as a Knowledge Transfer Network (KTN) which can be optimized to transform target embeddings to fit the source domain distribution. We perform an extensive evaluation of our method on 18 different transfer learning tasks on HGs where we compare against state-of-the-art domain adaptation baselines. Additionally, in order to understand which environments are ideal for transferring knowledge between different node types for HGs, we formulate a synthetic heterogeneous graph generator that allows us to study the sensitivity of these methods.

Our main contributions are:

- **Novel and practical problem definition:** To the best of our knowledge, KTN is the first zero-shot cross-type transfer learning method running on a heterogeneous graph — transfer knowledge across different node types within a heterogeneous graph.
- **Generality:** KTN is a principled approach analytically induced from the architecture of HGNNs, thus applicable to any HGNN models, showing up to $960\%$ performance improvement for zero-labeled node inference across 6 different HGNN models.
- **Effectiveness:** We show that KTN outperforms state-of-the-art domain adaptation methods, being up to $73.3\%$ higher in MRR on 18 different transfer learning tasks on HGs.
- **Sensitivity Analysis:** We provide a HG generator model to analyze how the node attribute and edge distributions of HGs affect the performance of KTN and other methods on the task.

## 2   Related Work

Various transfer learning problems have been defined on the graph domain. (21; 22; 38; 42) construct synthetic graphs from unstructured data and transfer knowledge over the graphs using GNNs. On

the other hand, (13; 14; 24; 37) focus on extracting knowledge from the existing graph structures. They pretrain a GNN model on a source graph and re-use the model on a target graph. While these methods focus on homogeneous graphs, (16; 40) transfer HGNNs across different HGs. However, none of them can be directly applied to our cross-type transfer learning problem running on a single HG. Here we cover two classes of learning approaches that are related to our problem. As HGNNs are the models to which our method can be applied, we cover them extensively in Section 3.

**Zero-shot domain adaptation (DA)**    transfers knowledge from a source domain with abundant labels to a target domain which lacks them. Zero-shot DA can be categorized into three groups — MMD-based methods, adversarial methods, and optimal-transport-based methods. MMD-based methods (18; 20; 29) minimize the maximum mean discrepancy (MMD) (11) between the mean embeddings of two distributions in reproducing kernel Hilbert space. Adversarial methods (9; 19) are motivated by theory in (2; 3) suggesting that a good cross-domain representation contains no discriminative information about the origin of the input. They learn domain-invariant features by a min-max game between the domain classifier and the feature extractor. Optimal transport-based methods (27) estimate the empirical Wasserstein distance (25) between two domains and minimizes the distance in an adversarial manner. All three categories rely on two networks — a feature extractor network and a task classifier network. Adversarial and OT-based methods use an additional domain classifier network. Due to the assumption that source and target domains have the same modality [2], the standard DA setting assumes identical feature extractors across domains. More descriptions can be found in Appendix A.9.

**Label propagation (LP)**    approaches (e.g., (45)) use message-passing to pass each node's label to its neighbors according to normalized edge weights. LP relies on only a graph's edges, and is therefore easily applied to a heterogeneous graph – labels are simply propagated across edges, regardless of type. In this paper we also evaluate a similarly-simple baseline, embedding propagation (EP). Similar to LP, EP recursively propagates source embeddings (computed using source labels) until they reach the target domain. EP exploits both node attribute information and the node adjacencies, but only uses the source node embeddings.

## 3    Preliminaries

In this section we review heterogeneous graphs and heterogeneous graph neural networks (HGNNs).

### 3.1    Heterogeneous graph

Heterogeneous graphs (HGs) are an important abstraction for modeling the relational data of multi-modal systems. Formally, a heterogeneous graph is defined as $\mathcal{G} = (\mathcal{V}, \mathcal{E}, \mathcal{T}, \mathcal{R})$ where the node set $\mathcal{V}$; the edge set $\mathcal{E}$ consisting of ordered tuples $e_{ij} := (i, j)$ with $i, j \in \mathcal{V}$, where $e_{ij} \in \mathcal{E}$ iff an edge exists from $i$ to $j$; the set of node types $\mathcal{T}$ with associated map $\tau : \mathcal{V} \mapsto \mathcal{T}$; the set of relation types $\mathcal{R}$ with associated map $\phi : \mathcal{E} \mapsto \mathcal{R}$. This flexible formulation allows directed, multi-type edges. We additionally assume the existence of a node attribute vector $x_i \in \mathcal{X}_{\tau(i)}$ for each $i \in \mathcal{V}$, where $\mathcal{X}_t$ is an attribute matrix specific to nodes of type $t$.

### 3.2    Heterogeneous Graph Neural Networks (HGNN)

Various HGNN models have been proposed (15; 26; 35; 41; 43). Fully-specified HGNN models have specialized parameters for each node type (15), edge type (26), and meta-path type (8) to most effectively utilize the complex relationships encoded in the HG data structure. In this paper, we use a flavor of HGNN known as a Heterogeneous Message-Passing Neural Network (HMPNN) as our base model on which to demonstrate KTN (though KTN can be implemented in almost any HGNN, as we show in experiments in Section 6). The HMPNN merely extends the standard MPNN (10) by specializing all transformation and message matrices in each node/edge type. With its generality, HMPNN is itself a base model for RGCN (26) and HGT (15), and is also widely used as a default HGNN model in popular GNN libraries (e.g., pyG (7), TF-GNN (6), DGL (34)).

---

[2]In our problem, source and target node types could have either (1) different distributions on the same attribute space or (2) entirely different attribute spaces

In a HMPNN, for any node $j$, the embedding of node $j$ at the $l$-th layer is obtained with the following generic formulation:

$$h_j^{(l)} = \textbf{Transform}^{(l)} \left( \textbf{Aggregate}^{(l)}(\mathcal{E}(j)) \right) \tag{1}$$

where $\mathcal{E}(j) = \{(i,j) \in \mathcal{E} : i, j \in \mathcal{V}\}$ denotes all the edges which connect (directionally) to $j$. The above operations typically involve type-specific parameters to exploit the inherent multiplicity of modalities in heterogeneous graphs. First, we define a linear **Message** function:

$$\textbf{Message}^{(l)}(i,j) = M_{\phi((i,j))}^{(l)} \cdot \left( h_i^{(l-1)} \parallel h_j^{(l-1)} \right) \tag{2}$$

where $M_r^{(l)}$ are the specific message passing parameters for each edge type $r \in \mathcal{R}$ and each of $L$ HMPNN layers. Then defining $\mathcal{E}_r(j)$ as the set of edges of type $r$ pointing to node $j$, the **Aggregate** function mean-pools messages by edge type, and concatenates:

$$\textbf{Aggregate}^{(l)}(\mathcal{E}(j)) = \mathop{\parallel}_{r \in \mathcal{R}} \frac{1}{|\mathcal{E}_r(j)|} \sum_{e \in \mathcal{E}_r(j)} \textbf{Message}^{(l)}(e) \tag{3}$$

Finally, the **Transform** function maps the message into a type-specific latent space:

$$\textbf{Transform}^{(l)}(j) = \alpha(W_{\tau(j)}^{(l)} \cdot \textbf{Aggregate}^{(l)}(\mathcal{E}(j))) \tag{4}$$

where $W_t^{(l)}$ are the specific transformation parameters for each node type $t \in \mathcal{T}$ and each of $L$ HMPNN layers. By stacking $L$ layers, HMPNN outputs highly contextualized final node representations, and the final node representations can be fed into another model to perform downstream heterogeneous network tasks, such as node classification or link prediction.

## 3.3 Problem definition

Using notations defined above, we formalize our novel transfer learning problem on HGs.

**Problem Definition 1. Zero-shot cross-type transfer learning running on a HG:**
*In a given heterogeneous graph $\mathcal{G} = (\mathcal{V}, \mathcal{E}, \mathcal{T}, \mathcal{R})$ with node attributes $\mathcal{X} = \cup_{t \in \mathcal{T}} \mathcal{X}_t$, assume node types* **s** *and* **t** *share a classification task $\{(i, y_i) : i \in \mathcal{V}_s, \mathcal{V}_t\}$. During the training phase, using labels $\{(i, y_i) : i \in \mathcal{V}_s\}$ only for source-type nodes, we train an HGNN model $f : f(\mathcal{G}, \mathcal{X}) = h_i$ to get node embeddings $h_i$ for all nodes $i \in \mathcal{V}$ and a classifier $g : g(h_i) = \hat{y}_i$ to predict labels $\hat{y}_i$ from the node embeddings $h_i$. During the test phase, our task is to predict labels $\{(j, y_j) : j \in \mathcal{V}_t\}$ of target-type nodes where none of labels of target-type nodes were used for training.*

# 4 Cross-Type Feature Extractor Transformations in HGNNs

We define $f_t : \mathcal{G} \mapsto \mathbb{R}^d$ to be the "feature extractor" of a HGNN, which is composed of parameters participating to map input node attributes of type $t$ into a shared feature space $\mathbb{R}^d$. In this section, we derive a strict transformation between feature extractors within a HMPNN. Specifically, for any two nodes $i, j$ with types $\tau(i) = s$ and $\tau(j) = t$, we derive an expression for $f_s$ in terms of $f_t$, and use that expression to inspire a trainable transfer learning module called KTN in the following section. For simplicity, throughout this section we ignore the activation $\alpha(\cdot)$ within the **Transform** function in Equation (4).

## 4.1 Feature extractors in HMPNNs

We first reason intuitively about the differences between $f_s$ and $f_t$ when $s \neq t$, using a toy heterogeneous graph shown in Figure 1(a). Consider nodes $v_1$ and $v_2$, noticing that $\tau(1) \neq \tau(2)$. Using HMPNN's equations (2)-(4) from Section 3.2, for any $l \in \{0, \dots, L-1\}$ we have

$$h_1^{(l)} = W_s^{(l)} \left[ M_{ss}^{(l)} \left( h_3^{(l-1)} \parallel h_1^{(l-1)} \right) \parallel M_{ts}^{(l)} \left( h_2^{(l-1)} \parallel h_1^{(l-1)} \right) \right] \tag{5}$$

$$h_2^{(l)} = W_t^{(l)} \left[ M_{st}^{(l)} \left( h_1^{(l-1)} \parallel h_2^{(l-1)} \right) \parallel M_{tt}^{(l)} \left( h_4^{(l-1)} \parallel h_2^{(l-1)} \right) \right] \tag{6}$$

where $h_j^{(0)} = x_j$. From these equations, we see that $h_1^{(l)}$ and $h_2^{(l)}$, which are features of different types, are extracted using *disjoint* sets of model parameters at $l$-th layer. In a 2-layer HMPNN, this creates unique gradient backpropagation paths between the two node types, as illustrated in Figures 1(b)-1(c). In other words, even though the same HMPNN is applied to node types $s$ and $t$, the feature extractors $f_s$ and $f_t$ have different computational paths. Therefore they project node features into different latent spaces, and have different update equations during training.

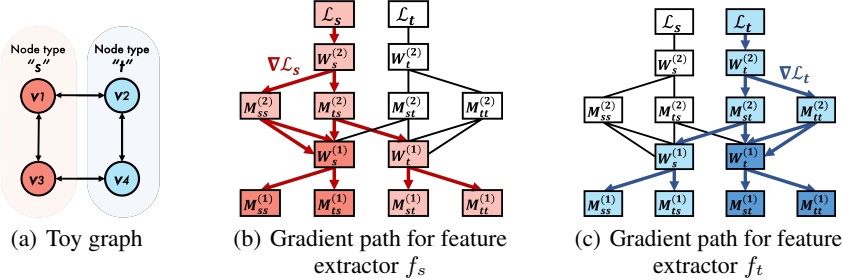

(a) Toy graph     (b) Gradient path for feature     (c) Gradient path for feature
                      extractor $f_s$                     extractor $f_t$

Figure 1: Illustration of a toy heterogeneous graph and the gradient paths for feature extractors $f_s$ and $f_t$. Colored arrows in figures (b) and (c) show that the same HGNN nonetheless produces different gradient paths for each feature extractor. Color density of each box in (b) and (c) is proportional to the degree of participation of the corresponding parameter in each feature extractor.

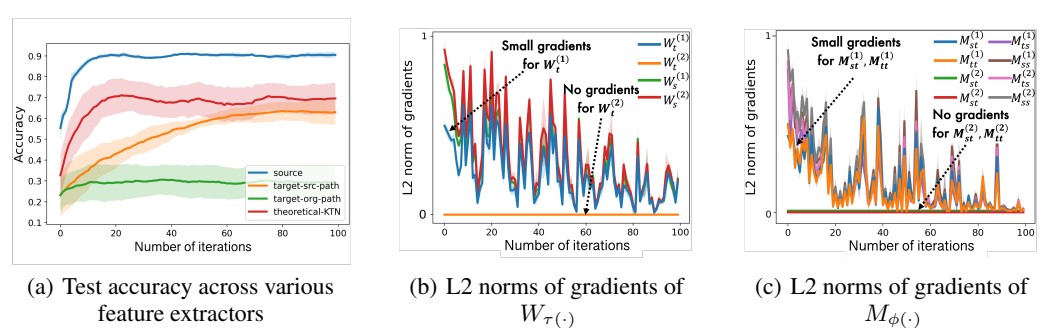

(a) Test accuracy across various       (b) L2 norms of gradients of       (c) L2 norms of gradients of
       feature extractors                        $W_{\tau(\cdot)}$                        $M_{\phi(\cdot)}$

Figure 2: HGNNs trained on a source domain underfit a target domain even on a "nice" heterogeneous graph. (a) Performance on the simulated heterogeneous graph for 4 kinds of feature extractors; (*source*: source extractor $f_s$ on source domain, *target-src-path*: source extractor $f_s$ on target domain, *target-org-path*: target extractor $f_t$ on target domain, and *theoretical-KTN*: target extractor $f_t$ on target domain using KTN.) (b-c) L2 norms of gradients of parameters $W_{\tau(\cdot)}$ and $M_{\phi(\cdot)}$ in HGNN models.

## 4.2   Empirical gap between $f_s$ and $f_t$

Here we study the experimental consequences of the above observation via simulation. We first construct a synthetic graph extending the 2-type graph in Figure 1(a) to have multiple nodes per-type, and multiple classes. To maximize the effects of having different feature extractors, we sample source and target nodes from the same feature distributions and each classes are well-separated in the both the graph and feature space (more details available in Appendix A.7.1).

On such a well-aligned heterogeneous graph, without considering the observation in Section 4.1, there may seem to be no need for domain adaptation from $f_t$ to $f_s$. However, when we train the HMPNN model solely on $s$-type nodes, as shown in Figure 2(a) we find the test accuracy for $s$-type nodes to be high (90%, blue line) and the test accuracy for $t$-type nodes to be quite low (25%, green line). Now if instead we make the $t$-type nodes use the source feature extractor $f_s$, much more transfer learning is possible (∼65%, orange line). This shows that the different feature extractors present in the HMPNN model result in the significant performance drop, and simply matching input data distributions can not solve the problem.

To analyze this phenomenon at the level of backpropagation, in Figures 2(b)-2(c) we show the magnitude of gradients passed to parameters of source and target node types. As illustrated in Figures 1(b)-1(c), we find that the final-layer **Transform** parameter $W_t^{(2)}$ for type-$t$ nodes have zero gradients (Figure 2(b)), and similarly for the final-layer **Message** parameters (Figure 2(c)). Additionally, those same parameters in the first-layer for $t$-type nodes have much smaller gradients than their $s$-type counterparts: $W_t^{(1)}$ (blue line in Figure 2(b)), $M_{st}^{(1)}$ and $M_{tt}^{(1)}$ (blue and orange lines in Figure 2(c)) appear below than other lines. This is because they contribute to $f_s$ less than $f_t$

This case study shows that even when an HGNN is trained on a relatively simple, balanced, and class-separated heterogeneous graph, a model trained only on the source domain node type cannot transfer to the target domain node type.

### 4.3 Relationship between feature extractors in HMPNNs

We show that a HMPNN model provides different feature extractors for each node type. However, still, $f_s$ and $f_t$ are built inside one HMPNN model and interchange intermediate feature embeddings with each other. Both $H_s^{(L)}$ and $H_t^{(L)}$ (the output of $f_s$ and $f_t$) are computed using the previous layer's intermediate embeddings $H_s^{(L-1)}, H_t^{(L-1)}$, and any other connected node type embeddings $H_x^{(L-1)}$ at the $L$-th HMPNN layer. Therefore $H_s^{(L)}$ and $H_t^{(L)}$ can be mathematically presented by each other using the $(L-1)$-th layer embeddings as connecting points, so do $f_s$ and $f_t$. Based on this intuition, we derive a strict transformation between $f_s$ and $f_t$, which will motivate the core domain adaptation component of our proposed KTN model.

**Theorem 1.** *Given a heterogeneous graph $\mathcal{G} = \{\mathcal{V}, \mathcal{E}, \mathcal{T}, \mathcal{R}\}$. For any layer $l > 0$, define the set of $(l-1)$-th layer HMPNN parameters as*

$$\mathcal{Q}^{(l-1)} = \{M_r^{(l-1)} : r \in \mathcal{R}\} \cup \{W_t^{(l-1)} : t \in \mathcal{T}\}. \tag{7}$$

*Let $A$ be the total $n \times n$ adjacency matrix. Then for any $s, t \in \mathcal{T}$ there exist matrices $A_{ts}^* = a_{ts}(A)$ and $Q_{ts}^* = q_{ts}(\mathcal{Q}^{(l-1)})$ such that*

$$H_s^{(l)} = A_{ts}^* H_t^{(l)} Q_{ts}^* \tag{8}$$

*where $a_{ts}(\cdot)$ and $q_{ts}(\cdot)$ are matrix functions that depend only on $s, t$.*

The full proof of Theorem 1 can be found in Appendix A.1. Notice that in Equation 8, $Q_{ts}^*$ acts as a macro-**Transform** operator that maps $H_t^{(L)}$ into the source domain, then $A_{ts}^*$ aggregates the mapped embeddings into $s$-type nodes. In other words, $Q_{ts}^*$ acts as a mapping matrix from the target domain to the source domain. To examine the implications, we run the same experiment as described in Section 4.2, while this time mapping the target features $H_t^{(L)}$ into the source domain by multiplying with $Q_{ts}^*$ in Equation 8 before passing over to a task classifier. We see via the red line in Figure 2(a) that, with this mapping, the accuracy in the target domain becomes much closer to the accuracy in the source domain ($\sim 70\%$). Thus, we use this theoretical transformation as a foundation for our trainable HGNN domain adaptation module, introduced in the following section.

### 4.4 Generalized cross-type transformations for HGNNs

In this section we showed that a HMPNN feature extractor on the (label-abundant) source node type can be expressed in terms of the (label-scarce) target node type feature extractor, and this transformation enables cross-type zero-shot learning for the target node type. As most HGNNs have distinct feature extractors for each node types (even single-layer HGNNs, which have specialized parameters for each node/edge attribute projection layer), they will suffer from the under-trained target embeddings phenomena we showed in Section 4.2. For instance, in the meta-path based MAGNN model (8), meta-paths directing toward the target node types are generally less engaged in the source node feature computation and thus receive smaller gradients. While we cannot derive the exact cross-type transformation for all possible HGNNs, the core intuition in the HMPNN theory holds, namely that $H_s^{(L)}$ and $H_t^{(L)}$ are both computed using the previous layer's intermediate embeddings (see Section 4.3) across all HGNN models. This observation allows us to extend our KTN and apply it to almost any HGNN. We illustrate this by applying KTN to 6 different HGNN models in Section 6, where we see greatly increased target-type accuracy.

## 5 KTN: Trainable Cross-Type Transfer Learning for HGNNs

Inspired by these derivations we introduce our primary contribution, *Knowledge Transfer Networks*. We begin by noting Equation 8 in Theorem 1 has a similar form to a single-layer graph convolutional network (17) with a deterministic transformation matrix ($Q_{ts}^*$) and a combination of adjacency matrices directing from target node type $t$ to source node type $s$ ($A_{ts}^*$). Instead of hand-computing the mapping function $Q_{ts}^*$ for arbitrary HGs and HGNNs (which would be intractable), we *learn* the mapping function by modelling Equation 8 as a trainable graph convolutional network, named the

---
**Algorithm 1** Training step on a source domain
---
**Require:** heterogeneous graph $\mathcal{G} = (\mathcal{V}, \mathcal{E}, \mathcal{T}, \mathcal{R})$, node feature matrices $\mathcal{X}$, source node type $s$, target node type $t$, adjacency matrix $A_{ts}$, source node label matrix $\mathcal{Y}_s$.
**Ensure:** HGNN $\mathbf{f}$, classifier $\mathbf{g}$, KTN $\mathbf{t}_{\text{KTN}}$
1: $H_s^{(L)}, H_t^{(L)} = \mathbf{f}(\mathcal{G}, H^{(0)} = \mathcal{X})$
2: $H_t^* = \mathbf{t}_{KTN}(H_t^{(L)}) = A_{ts} H_t^{(L)} T_{ts}$
3: $\mathcal{L}_{\text{KTN}} = \left\| H_s^{(L)} - H_t^* \right\|_2$
4: $\mathcal{L} = \mathcal{L}_{\text{CL}}(\mathbf{g}(H_s^{(L)}), \mathcal{Y}_s) + \lambda \mathcal{L}_{\text{KTN}}$
5: Update $\mathbf{f}, \mathbf{g}, \mathbf{t}$ using $\nabla \mathcal{L}$
---

---
**Algorithm 2** Test step on a target domain
---
**Require:** pretrained HGNN $\mathbf{f}$, classifier $\mathbf{g}$, KTN $\mathbf{t}_{\text{KTN}}$
**Ensure:** target node label matrix $\mathcal{Y}_t$
1: $H_t^{(L)} = \mathbf{f}(\mathcal{G}, H^{(0)} = \mathcal{X})$
2: $H_t^* = H_t^{(L)} T_{ts}$
3: **return** $\mathbf{g}(H_t^*)$
---

Knowledge Transfer Network, $\mathbf{t}_{\text{KTN}}(\cdot)$. KTN replaces $Q_{ts}^*$ and $A_{ts}^*$ in Equation 8 as follows:

$$\mathbf{t}_{\text{KTN}}(H_t^{(L)}) = A_{ts} H_t^{(L)} T_{ts} \tag{9}$$

$$\mathcal{L}_{\text{KTN}} = \left\| H_s^{(L)} - \mathbf{t}_{\text{KTN}}(H_t^{(L)}) \right\|_2 \tag{10}$$

where $A_{ts}$ is an adjacency matrix from node type $t$ to $s$, and $T_{ts}$ is a trainable transformation matrix. By minimizing $\mathcal{L}_{\text{KTN}}$, $T_{ts}$ is optimized to a mapping function of the target domain into the source domain.

## 5.1 Algorithm

We minimize a classification loss $\mathcal{L}_{\text{CL}}$ and a transfer loss $\mathcal{L}_{\text{KTN}}$ jointly with regard to a HGNN model $\mathbf{f}$, a classifier $\mathbf{g}$, and a knowledge transfer network $\mathbf{t}_{\text{KTN}}$ as follows:

$$\min_{\mathbf{f}, \, \mathbf{g}, \, \mathbf{t}_{\text{KTN}}} \mathcal{L}_{\text{CL}}(\mathbf{g}(\mathbf{f}(\mathcal{G}, \mathcal{X})_s), \mathcal{Y}_s) + \lambda \left\| \mathbf{f}(\mathcal{G}, \mathcal{X})_s - \mathbf{t}_{\text{KTN}}(\mathbf{f}(\mathcal{G}, \mathcal{X})_t) \right\|_2$$

where $\lambda$ is a hyperparameter regulating the effect of $\mathcal{L}_{\text{KTN}}$; and $\mathbf{f}(\mathcal{G}, \mathcal{X})_s$ and $\mathbf{f}(\mathcal{G}, \mathcal{X})_t$ denote $H_s^{(L)}$ and $H_t^{(L)}$, respectively. Algorithm 1 describes a training step on the source domain. After computing the node embeddings $H_s^{(L)}$ and $H_t^{(L)}$, we map $H_t^{(L)}$ to the source domain using $\mathbf{t}_{\text{KTN}}$ and compute $\mathcal{L}_{\text{KTN}}$. Then, we update the models using gradients of $\mathcal{L}_{\text{CL}}$ (computed using only source labels) and $\mathcal{L}_{\text{KTN}}$. Algorithm 2 describes the test phase on the target domain. After we get node embeddings $H_t^{(L)}$ from the trained HGNN model, we map $H_t^{(L)}$ into the source domain using the trained transformation matrix $T_{ts}$. Finally we pass the transformed target embeddings $H_t^*$ into the classifier which was trained on the source domain.

**Indirect Connections** We note that in practice, the source and target node types can be indirectly connected in HGs via other node types (i.e., there is no $A_{ts}$). Appendix A.2 describes how we can easily extend KTN to cover domain adaption scenarios in this case.

## 6 Experiments

### 6.1 Datasets

**Open Academic Graph (OAG).** A dataset introduced in (44) composed of five types of nodes: papers (P), authors (A), institutions (I), venues (V), fields (F) and their corresponding relationships. Paper and author nodes have text-based attributes, while institution, venue, and field nodes have text- and graph structure-based attributes. Paper, author, and venue nodes are labeled with research fields in two hierarchical levels, L1 and L2. We construct three field-specific subgraphs from OAG: computer science, computer networks, and machine learning academic graphs.

Table 1: **Open Academic Graph on Computer Science field**. The *gain* column shows the relative gain of our method over using no domain adaptation (*Base* column). *o.o.m* denotes *out-of-memory* errors.

| Task | Metric | Base | DAN | JAN | DANN | CDAN | CDAN-E | WDGRL | LP | EP | KTN (gain) |
|------|--------|------|-----|-----|------|------|--------|-------|----|----|-----------|
| P-A (L1) | NDCG | 0.399 | 0.452 | 0.405 | 0.292 | 0.262 | 0.261 | 0.260 | 0.178 | 0.425 | **0.623 (56%)** |
|  | MRR | 0.297 | 0.361 | 0.314 | 0.179 | 0.129 | 0.111 | 0.138 | 0.041 | 0.363 | **0.629 (112%)** |
| A-P (L1) | NDCG | 0.401 | 0.566 | 0.598 | 0.294 | 0.364 | 0.246 | 0.195 | 0.153 | 0.557 | **0.733 (83%)** |
|  | MRR | 0.318 | 0.508 | 0.544 | 0.229 | 0.270 | 0.090 | 0.047 | 0.022 | 0.507 | **0.711 (123%)** |
| A-V (L1) | NDCG | 0.459 | 0.457 | 0.470 | 0.382 | 0.346 | 0.359 | 0.403 | 0.207 | 0.461 | **0.671 (46%)** |
|  | MRR | 0.364 | 0.413 | 0.458 | 0.341 | 0.205 | 0.253 | 0.327 | 0.011 | 0.389 | **0.698 (92%)** |
| V-A (L1) | NDCG | 0.283 | 0.443 | 0.435 | 0.242 | 0.372 | 0.418 | 0.272 | 0.153 | 0.154 | **0.584 (107%)** |
|  | MRR | 0.133 | 0.365 | 0.345 | 0.094 | 0.241 | 0.444 | 0.144 | 0.006 | 0.006 | **0.586 (340%)** |
| P-A (L2) | NDCG | 0.229 | 0.230 | o.o.m | 0.239 | o.o.m | o.o.m | 0.168 | o.o.m | 0.215 | **0.282 (23%)** |
|  | MRR | 0.121 | 0.118 | o.o.m | 0.140 | o.o.m | o.o.m | 0.020 | o.o.m | 0.143 | **0.2248 (86%)** |
| A-P (L2) | NDCG | 0.197 | 0.162 | o.o.m | 0.204 | 0.158 | 0.161 | 0.132 | o.o.m | 0.208 | **0.287 (46%)** |
|  | MRR | 0.095 | 0.052 | o.o.m | 0.106 | 0.032 | 0.045 | 0.017 | o.o.m | 0.132 | **0.242 (155%)** |
| A-V (L2) | NDCG | 0.347 | 0.329 | 0.295 | 0.325 | 0.288 | 0.273 | 0.289 | o.o.m | 0.297 | **0.402 (16%)** |
|  | MRR | 0.310 | 0.296 | 0.198 | 0.223 | 0.128 | 0.097 | 0.110 | o.o.m | 0.227 | **0.399 (29%)** |
| V-A (L2) | NDCG | 0.235 | 0.249 | 0.251 | 0.214 | 0.197 | 0.205 | 0.217 | o.o.m | 0.119 | **0.252 (7%)** |
|  | MRR | 0.129 | 0.157 | 0.161 | 0.090 | 0.044 | 0.068 | 0.085 | o.o.m | 0.000 | **0.166 (28%)** |

Table 2: **PubMed graph**. The *gain* column shows the relative gain over using *Base* column.

| Task | Metric | Base | DAN | JAN | DANN | CDAN | CDAN-E | WDGRL | LP | EP | KTN (gain) |
|------|--------|------|-----|-----|------|------|--------|-------|----|----|-----------|
| D-G | NDCG | 0.587 | 0.629 | 0.615 | 0.614 | 0.624 | 0.646 | 0.604 | 0.601 | 0.571 | **0.700 (19%)** |
|  | MRR | 0.372 | 0.425 | 0.414 | 0.397 | 0.428 | 0.443 | 0.388 | 0.389 | 0.336 | **0.499 (34%)** |
| G-D | NDCG | 0.596 | 0.599 | 0.577 | 0.599 | 0.581 | 0.606 | 0.578 | 0.576 | 0.580 | **0.662 (11%)** |
|  | MRR | 0.354 | 0.362 | 0.332 | 0.356 | 0.337 | 0.362 | 0.340 | 0.351 | 0.353 | **0.445 (26%)** |

**PubMed.**(39) A network composed of of four types of nodes: genes (G), diseases (D), chemicals (C), and species (S), and their corresponding relationships. Gene and chemical nodes have graph structure-based attributes, while disease and species nodes have text-based attributes. Each gene and disease is labeled with a set of diseases they belong to or cause.

**Synthetic heterogeneous graphs.** We generate stochastic block models (1) with multiple node/edge types. We label each node types with the same set of classes. Then we control feature/edge distributions within/between node types by manipulating feature/edge signal-to-noise ratio within/between classes. A complete definition of the generative model is given in Appendix A.7.

## 6.2 Baselines

We compare KTN with two MMD-based DA methods (DAN (18), JAN (20)), three adversarial DA methods (DANN (9), CDAN (19), CDAN-E (19)), one optimal transport-based method (WD-GRL (27)), and two traditional graph mining methods (LP and EP (45)). For KTN and DA methods, we use HMPNN as their base HGNN model. More information of each method is in Appendix A.9.

## 6.3 Zero-shot transfer learning

We run 18 different zero-shot transfer learning tasks across three OAG and PubMed graphs. We run each experiment 3 times and report the average value. Due to the space limitation, we report the standard deviations and results on OAG-computer networks and OAG-machine learning in Appendix A.3. Each heterogeneous graph has the same node classification task for both source and target node types. During training, we are given 1) the heterogeneous graph structure information (i.e., adjacency matrices), 2) input node attribute matrices for all node types, and 3) labels on source-type nodes for the classification task. During the test phase, we predict labels on target-type nodes for the same classification task. The performance is evaluated by NDCG and MRR — widely adopted ranking metrics (14; 15).

In Tables 1 and 2, our proposed method KTN consistently outperforms all baselines on all tasks and graphs by up to 73.3% higher in MRR (P-A(L1) task in OAG-CS, Table 1). When we compare with the base accuracy using the model pretrained on the source domain without any domain adaptation (3rd column, *Base*), the results are even more impressive. We see our method KTN provides relative gains of up to 340% higher MRR without using any labels from the target domain. These results show the clear effectiveness of KTN on zero-shot transfer learning tasks on a heterogeneous graph. We mention that venue and author node types are not directly connected in the OAG graphs (Figure 5(b) in Appendix), but KTN successfully transfer knowledge by passing intermediate node types.

Table 3: **KTN on different HGNN models.** The *Source* column shows accuracy on for source node types. *Base* and *KTN* columns show accuracy for target node types without/with using KTN, respectively. The *Gain* column shows the relative gain of our method over using no domain adaptation.

| HGNN type | Metric | P-A (L1) | | | | A-P (L1) | | | |
|---|---|---|---|---|---|---|---|---|---|
| | | Source | Base | KTN | Gain | Source | Base | KTN | Gain |
| R-GCN | NDCG | 0.765 | 0.337 | 0.577 | **71.12%** | 0.648 | 0.388 | 0.647 | **66.82%** |
| | MRR | 0.757 | 0.236 | 0.587 | **148.73%** | 0.623 | 0.270 | 0.611 | **126.18%** |
| HAN | NDCG | 0.476 | 0.179 | 0.520 | **190.56%** | 0.515 | 0.182 | 0.512 | **181.33%** |
| | MRR | 0.416 | 0.047 | 0.497 | **960.55%** | 0.509 | 0.055 | 0.527 | **850.90%** |
| HGT | NDCG | 0.757 | 0.294 | 0.574 | **95.07%** | 0.670 | 0.283 | 0.581 | **104.83%** |
| | MRR | 0.749 | 0.178 | 0.563 | **216.17%** | 0.670 | 0.149 | 0.565 | **279.52%** |
| MAGNN | NDCG | 0.657 | 0.463 | 0.574 | **24.01%** | 0.676 | 0.557 | 0.622 | **11.68%** |
| | MRR | 0.631 | 0.378 | 0.556 | **47.33%** | 0.680 | 0.509 | 0.592 | **16.14%** |
| MPNN | NDCG | 0.602 | 0.443 | 0.590 | **33.11%** | 0.646 | 0.307 | 0.621 | **101.92%** |
| | MRR | 0.572 | 0.319 | 0.575 | **80.10%** | 0.660 | 0.145 | 0.595 | **311.42%** |
| HMPNN | NDCG | 0.789 | 0.399 | 0.623 | **56.14%** | 0.671 | 0.401 | 0.733 | **82.88%** |
| | MRR | 0.777 | 0.297 | 0.629 | **111.86%** | 0.661 | 0.318 | 0.711 | **123.30%** |

**Baseline Performance.** Among baselines, MMD-based models (DAN and JAN) outperform adversarial-based methods (DANN, CDAN, and CDAN-E) and optimal transport-based method (WDGRL), unlike results reported in (19; 27). These reversed results are a consequence of HGNN's unique feature extractors for each domains. Adversarial- and optimal transport-based methods define separate losses for source and target feature extractors (which are not separated in their shared feature extractor assumption), resulting in divergent gradients between different feature extractors and poor domain adaption performance. This shows again the importance of considering different feature extractors in HGNNs. More analysis can be found in Appendix A.4.

## 6.4 Generality of KTN

Here, we use 6 different HGNN models, R-GCN (26), HAN (35), HGT (15), MAGNN (8), MPNN (10), and HMPNN. MPNN maps all node types to the shared embedding space using projection matrices at the beginning then applies MPNN layers designed for homogeneous graphs. In Table 3, KTN improves accuracy on the target nodes across all HGNN models by up to $960\%$. This shows the strong generality of KTN. More results and analysis can be found in Appendix A.5.

## 6.5 Sensitivity analysis

Using our synthetic heterogeneous graph generator, we generate non-trivial 2-type heterogeneous graphs to examine how the feature and edge distributions affect the performance of KTN and other baselines. We generate a *range* of test-case scenarios by manipulating (1) signal-to-noise ratio $\sigma_e$ of within-class edge distributions and (2) signal-to-noise ratio $\sigma_f$ of within-class feature distributions across all of the (a) source-source ($s \leftrightarrow s$), (b) target-target ($t \leftrightarrow t$), and (c) source-target ($s \leftrightarrow t$) relationships.

For instance, in Figure 3, for each edge type ($s \leftrightarrow s$, $t \leftrightarrow t$, and $s \leftrightarrow t$, differentiated by colors), there are two different types of edges, edges within the same class (plain line) and edges across different classes (dotted line). For each edge type, we manipulate $\sigma_e$ by changing the ratio of within-class and cross-class edges, and $\sigma_f$ by diverging feature distributions between classes. Thus there will be 6 signal-to-noise ratios in total. A higher signal-to-noise ratio for a particular data dimension (edges or features) across a particular relationship $r \in \{s \leftrightarrow s, \ t \leftrightarrow t, \ s \leftrightarrow t\}$ means that

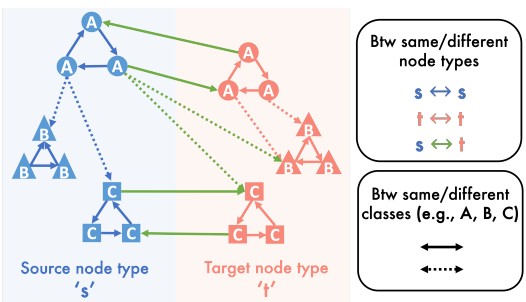

Figure 3: Synthetic HG generator

classes are more *separable* in that data dimension, when comparing within $r$, and hence easier for HGNNs. Note that while tuning one $\sigma_{(\cdot)}$ on the range $[1.0, 10.0]$, the remaining five $\sigma_{(\cdot)}$ are held at 10.0. Additionally, we vary $\sigma_{(\cdot)}$ across two scenarios: (I) "easy": source and target node types have same number of clusters and same feature dimensions, (II) "hard" source and target node types have different number of clusters and feature dimensions. Note that clusters and classes are different concepts in this experiment; several clusters could have the same class label.

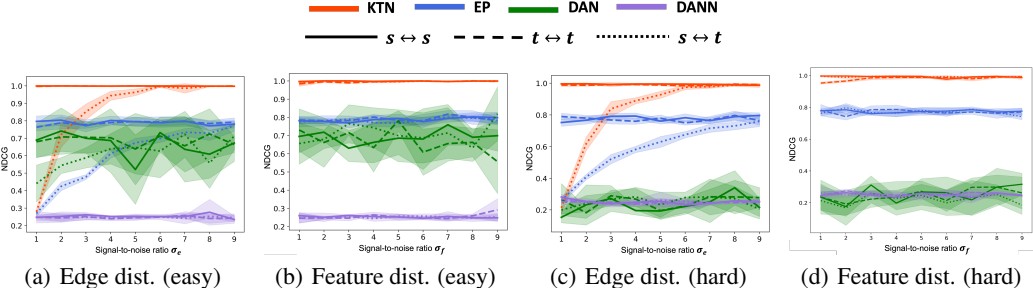

(a) Edge dist. (easy)  (b) Feature dist. (easy)  (c) Edge dist. (hard)  (d) Feature dist. (hard)

Figure 4: Effects of edge and feature distributions across classes and types in heterogeneous graphs.

Figures 4(a) and 4(c) show results from changing $\sigma_e$ across the three relation types. We see that KTN is affected only by $\sigma_e$ across the $s \leftrightarrow t$ (cross-types) relationship, which accords with our theory, since KTN exploits the between-type adjacency matrix. Surprisingly, as seen in Figures 4(b) and 4(d), we do not find a similar dependence of KTN on $\sigma_f$, which shows that KTN is robust by learning purely from edge homophily in the absence of feature homophily. Regarding the performance of other baselines, EP shows similar tendencies as KTN— only affected by cross-type $\sigma_e$ — because EP also relies on cross-type propagation along edges. However, its accuracy is bounded above due to the fact that it does not exploit the (unlabelled) target features. DAN and DANN, which do not exploit cross-type edges, are not affected by cross-type $\sigma_e$. However, they show either low or unstable performance across different scenarios. DAN shows especially poor performance in the "hard" scenarios (Figure 4(c) and 4(d)), failing to deal with different feature spaces for source and target domains.

## 7  Conclusion

In this work, we proposed the first cross-type zero-shot transfer learning method for heterogeneous graphs. Our method, Knowledge Transfer Networks (KTN) for Heterogeneous Graph Neural Networks, transfers knowledge from *label-abundant* node types to *label-scarce* node types. We illustrate KTN handily improves HGNN performances up to $960\%$ for zero-labeled node types across 6 different HGNN models and outperforms many challenging baselines up to $73\%$ higher in MRR. Future work in the area includes filtering noisy edges between source and target domains and making KTN more robust and less dependent on structure of given noisy heterogeneous graphs.

**Limitation Statement**  Our transfer learning method is limited to node types sharing the same task (i.e., the same classifier). We plan to extend our work to transfer knowledge between different tasks running on different node types on a heterogeneous graph.

**Societal Impact Statement**  KTN allows organizations to learn better from their own graph data, leveraging its structure without requiring external information. We believe this has a number of positive applications (preserving model quality without needing extra datasets). However like all modeling improvements, its true impact depends on what modeling tasks the technique is applied to.

## 8  Acknowledgement

MY gratefully acknowledges support from Amazon Graduate Research Fellowship. GPUs are partially supported by AWS Cloud Credit for Research program.

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
