# A  Appendix

## A.1  Proof of Theorem 1

In this proof, we adopt a simplified version of our message-passing function that ignores the skip-connection:

$$\textbf{Message}^{(l)}(i,j) = M^{(l)}_{\phi(i,j)} h^{(j)}_i. \tag{11}$$

The HGNN trained in the experimental results shown in Figure 2 also does not use skip-connections and hence represents a theoretically-exact KTN component. In the real experiments, we use (1) skip-connections, exploiting their usual benefits (12), and (2) the trainable version of KTN.

*Proof.* Without loss of generality, we prove the result for the case where $\mathcal{R} = \{(s,t) : s, t \in \mathcal{T}\}$, meaning the type of an edge is identified with the (ordered) types of the neighbor nodes. In other words, there is only one edge modality possible, such as a social networks with multiple node types (e.g. "users", "groups") but only one edge modality ("friendship"). In the case of multiple edge modalities (e.g. "friendship" and "message"), the result is extended trivially (through with more algebraically-dense forms of $a_{ts}$ and $q_{ts}$).

Throughout this proof, we use the following notation for the set of all $j$-adjacent edges of relation type $r$:

$$\mathcal{E}_r(j) := \{(i,j) : i \in \mathcal{V}, (i,j) = r\}. \tag{12}$$

We write $A_{x_1 x_2}$ to denote the sub-matrix of the total $n_{x_1} \times n_{x_2}$ adjacency matrix $A$ corresponding to node types $x_1, x_2 \in \mathcal{T}$, and $\bar{A}_{x_1 x_2}$ to denote the same matrix divided by its column sum. $H^{(l)}_x$ is the (row-wise) $n_x \times d_l$ embedding matrix of $x$-type nodes at layer $l$.

We first compute the *l-th* output $g^{(l)}_j$ of the **Aggregate** step defined for HGNNs in Equation 3, for any node $j \in \mathcal{V}$ such that $\tau(j) = s$. The output of **Aggregate** is a concatenation of edge-type-specific aggregations (see Equation 3). Note that at most $T = |\mathcal{T}|$ elements of this concatenation are non-zero, since the node $j$ only participates in $T$ out of $T^2$ relation types in $\mathcal{R}$. Thus we can write $g^{(l)}_j$ as

$$
\begin{aligned}
g^{(l)}_j &= \mathop{\|}_{r \in \mathcal{R}} \frac{1}{|\mathcal{E}_r(j)|} \sum_{e \in \mathcal{E}_r(j)} \textbf{Message}^{(l)}(e) \\
&= \mathop{\|}_{x \in \mathcal{T}} \frac{1}{|\mathcal{E}_{xs}(j)|} \sum_{e \in \mathcal{E}_{xs}(j)} \textbf{Message}^{(l)}(e) \\
&= \mathop{\|}_{x \in \mathcal{T}} \frac{1}{|\mathcal{E}_{xs}(j)|} \sum_{(i,j) \in \mathcal{E}_{xs}(j)} M^{(l)}_{xs} h^{(l-1)}_i \\
&= \mathop{\|}_{x \in \mathcal{T}} \frac{1}{|\mathcal{E}_{xs}(j)|} M^{(l)}_{xs} \sum_{(i,j) \in \mathcal{E}_{xs}(j)} h^{(l-1)}_i \\
&= \mathop{\|}_{x \in \mathcal{T}} M^{(l)}_{xs} \left( H^{(l-1)}_x \right)' \bar{A}^{(j)}_{xs},
\end{aligned}
$$

where $\bar{A}^{(j)}_{xs}$ denotes the *j-th* column of $\bar{A}_{xs}$. Notice that

$$h^{(l)}_j = \textbf{Transform}^{(l)}(j) = W^{(l)}_s g^{(l)}_j, \tag{13}$$

and (again) at most $T$ elements of the concatenation $g^{(l)}_j$ are non-zero. Therefore let $W^{(l)}_{xs}$ be the columns of $W^{(l)}_s$ that select the concatenated element of $g^{(l)}_j$ corresponding to node type $x$. Then we can write

$$h^{(l)}_j = \sum_{x \in \mathcal{T}} W^{(l)}_{xs} M^{(l)}_{xs} \left( H^{(l-1)}_x \right)' \bar{A}^{(j)}_{xs}. \tag{14}$$

**Algorithm 3** Training step for one minibatch (indirect version)
___
**Require:** heterogeneous graph $\mathcal{G} = (\mathcal{V}, \mathcal{E}, \mathcal{T}, \mathcal{R})$, node feature matrices $X$, adjacency matrices $A_{xy}$ where $\forall (x, y) \in \mathcal{R}$, source node type $s$, target node type $t$, source node label matrix $Y_s$.
**Ensure:** HGNN **f**, classifier **g**, KTN $\mathbf{t}_{\text{KTN}}$
1: $H_s^{(L)}, H_t^{(L)} = \mathbf{f}(H^{(0)} = X, \mathcal{G}), H_t^* = \mathbf{0}$
2: **for** each meta-path $p = t \to s$ **do**
3:   $x = t, Z = H_t^{(L)}$
4:   **for** each node type $y \in p$ **do**
5:    $Z = A_{xy} Z T_{xy}$
6:    $x = y$
7:   **end for**
8:   $H_t^* = H_t^* + Z$
9: **end for**
10: $\mathcal{L}_{\text{KTN}} = \left\| H_s^{(L)} - H_t^* \right\|_2$
11: $\mathcal{L} = \mathcal{L}_{\text{CL}}(\mathbf{g}(H_s^{(L)}), Y_s) + \lambda \mathcal{L}_{\text{KTN}}$
12: Update **f**, **g**, $\mathbf{t}_{\text{KTN}}$ using $\nabla \mathcal{L}$
___

**Algorithm 4** Test step for a target domain (indirect version)
___
**Require:** pretrained HGNN **f**, classifier **g**, KTN $\mathbf{t}_{\text{KTN}}$
**Ensure:** target node label matrix $Y_t$
1: $H_t^{(L)} = \mathbf{f}(H^{(0)} = X, \mathcal{G}), H_t^* = \mathbf{0}$
2: **for** each meta-path $p = t \to s$ **do**
3:   $x = t, Z = H_t^{(L)}$
4:   **for** each node type $y \in p$ **do**
5:    $X = Z T_{xy}$
6:    $x = y$
7:   **end for**
8:   $H_t^* = H_t^* + Z$
9: **end for**
10: **return** $\mathbf{g}(H_t^*)$
___

Defining the operator $Q_{xs}^{(l)} := \left( W_{xs}^{(l)} M_{xs}^{(l)} \right)'$, this implies that

$$H_s^{(l)} = \sum_{x \in \mathcal{T}} \bar{A}_{xs} H_x^{(l-1)} Q_{xs}^{(l-1)}$$

$$= [\bar{A}_{x_1 s}, \dots, \bar{A}_{x_T s}] \begin{bmatrix} H_{x_1}^{(l-1)} & 0 & 0 \\ 0 & \dots & 0 \\ 0 & 0 & H_{x_T}^{(l-1)} \end{bmatrix} \begin{bmatrix} Q_{x_1 s}^{(l-1)} \\ \dots \\ Q_{x_T s}^{(l-1)} \end{bmatrix}$$

$$= \bar{A}_{\cdot s} H_{\cdot}^{(l-1)} Q_{\cdot s}^{(l-1)}$$

Similarly we have $H_t^{(l)} = \bar{A}_{\cdot t} H_{\cdot}^{(l-1)} Q_{\cdot t}^{(l-1)}$. Since $H_s^{(l)}$ and $H_t^{(l)}$ share the term $H_{\cdot}^{(l-1)}$, we can write

$$H_s^{(l)} = \bar{A}_{\cdot s} \bar{A}_{\cdot t}^{-1} H_t^{(l)} (Q_{\cdot t}^{(l-1)})^{-1} Q_{\cdot s}^{(l-1)}, \qquad (15)$$

where $X^{-1}$ denotes the pseudo-inverse.          ∎

## A.2   Indirectly Connected Source and Target Node Types

When source and target node types are indirectly connected by another node type $x$, we can simply extend $\mathbf{t}_{\text{KTN}}(H_t^{(L)})$ to $(A_{xs}(A_{tx} H_t^{(L)} T_{tx}) T_{xs})$ where $T_{tx} T_{xs}$ becomes a mapping function from target to source domains. Algorithms 3 and 4 show how to extend KTN. For every step $(x \to y)$ in a meta-path $(t \to \cdots \to s)$ connecting target node type $t$ to source node type $s$, we define a transformation matrix $T_{xy}$, run a convolution operation with an adjacency matrix $A_{xy}$, then map the transformed embedding to the source domain. We run the same process for all meta-paths connecting from target node type $t$ to source node type $s$, and sum up them to match with the source embeddings. In the test phase, we run the same process to get the transformed target embeddings, but this time, without adjacency matrices. We run Algorithm 3 and 4 for transfer learning tasks between author and venue nodes which are indirectly connected by paper nodes in OAG graphs (Figure 5(b)). As shown

Table 4: **Open Academic Graph on Computer Science field**. The *gain* column shows the relative gain of our method over using no domain adaptation (*Base* column). *o.o.m* denotes *out-of-memory* errors.

| Task | Metric | Base | DAN | JAN | DANN | CDAN | CDAN-E | WDGRL | LP | EP | KTN (gain%) |
|---|---|---|---|---|---|---|---|---|---|---|---|
| P-A (L1) | NDCG | 0.399 | 0.452 | 0.405 | 0.292 | 0.262 | 0.261 | 0.26 | 0.178 | 0.425 | **0.623 (56)** |
| | std | 0.010 | 0.012 | 0.032 | 0.009 | 0.021 | 0.014 | 0.021 | 0.000 | 0.006 | 0.004 |
| | MRR | 0.297 | 0.361 | 0.314 | 0.179 | 0.129 | 0.111 | 0.138 | 0.041 | 0.363 | **0.629 (112)** |
| | std | 0.024 | 0.006 | 0.041 | 0.011 | 0.032 | 0.031 | 0.033 | 0.000 | 0.005 | 0.004 |
| A-P (L1) | NDCG | 0.401 | 0.566 | 0.598 | 0.294 | 0.364 | 0.246 | 0.195 | 0.153 | 0.557 | **0.733 (83)** |
| | std | 0.003 | 0.012 | 0.014 | 0.034 | 0.049 | 0.046 | 0.025 | 0.000 | 0.002 | 0.007 |
| | MRR | 0.318 | 0.508 | 0.544 | 0.229 | 0.27 | 0.09 | 0.047 | 0.022 | 0.507 | **0.711 (123)** |
| | std | 0.001 | 0.029 | 0.028 | 0.093 | 0.117 | 0.037 | 0.029 | 0.000 | 0.003 | 0.009 |
| A-V (L1) | NDCG | 0.459 | 0.457 | 0.47 | 0.382 | 0.346 | 0.359 | 0.403 | 0.207 | 0.461 | **0.671 (46)** |
| | std | 0.030 | 0.033 | 0.036 | 0.015 | 0.029 | 0.109 | 0.024 | 0.000 | 0.002 | 0.004 |
| | MRR | 0.364 | 0.413 | 0.458 | 0.341 | 0.205 | 0.253 | 0.327 | 0.011 | 0.389 | **0.698 (92)** |
| | std | 0.079 | 0.08 | 0.093 | 0.05 | 0.098 | 0.143 | 0.044 | 0.000 | 0.004 | 0.003 |
| V-A (L1) | NDCG | 0.283 | 0.443 | 0.435 | 0.242 | 0.372 | 0.418 | 0.272 | 0.153 | 0.154 | **0.584 (107)** |
| | std | 0.045 | 0.012 | 0.007 | 0.004 | 0.048 | 0.039 | 0.004 | 0.000 | 0.006 | 0.005 |
| | MRR | 0.133 | 0.365 | 0.345 | 0.094 | 0.241 | 0.444 | 0.144 | 0.006 | 0.006 | **0.586 (340)** |
| | std | 0.074 | 0.027 | 0.017 | 0.011 | 0.103 | 0.115 | 0.018 | | 0.007 | 0.010 |
| P-A (L2) | NDCG | 0.229 | 0.23 | o.o.m | 0.239 | o.o.m | o.o.m | 0.168 | o.o.m | 0.215 | **0.282 (23)** |
| | std | 0.005 | 0.003 | - | 0.006 | - | - | 0.007 | - | 0.004 | 0.002 |
| | MRR | 0.121 | 0.118 | o.o.m | 0.14 | o.o.m | o.o.m | 0.02 | o.o.m | 0.143 | **0.2248 (86)** |
| | std | 0.019 | 0.004 | - | 0.01 | - | - | 0.006 | - | 0.003 | 0.003 |
| A-P (L2) | NDCG | 0.197 | 0.162 | o.o.m | 0.204 | 0.158 | 0.161 | 0.132 | o.o.m | 0.208 | **0.287 (46)** |
| | std | 0.006 | 0.009 | - | 0.006 | 0.019 | 0.022 | 0.012 | - | 0.004 | 0.001 |
| | MRR | 0.095 | 0.052 | o.o.m | 0.106 | 0.032 | 0.045 | 0.017 | o.o.m | 0.132 | **0.242 (155)** |
| | std | 0.009 | 0.022 | - | 0.016 | 0.018 | 0.027 | 0.008 | - | 0.005 | 0.002 |
| A-V (L2) | NDCG | 0.347 | 0.329 | 0.295 | 0.325 | 0.288 | 0.273 | 0.289 | o.o.m | 0.297 | **0.402 (16)** |
| | std | 0.003 | 0.034 | 0.014 | 0.013 | 0.011 | 0.058 | 0.011 | - | 0.002 | 0.003 |
| | MRR | 0.310 | 0.296 | 0.198 | 0.223 | 0.128 | 0.097 | 0.11 | o.o.m | 0.227 | **0.399 (29)** |
| | std | 0.004 | 0.109 | 0.047 | 0.065 | 0.003 | 0.096 | 0.034 | - | 0.001 | 0.015 |
| V-A (L2) | NDCG | 0.235 | 0.249 | 0.251 | 0.214 | 0.197 | 0.205 | 0.217 | o.o.m | 0.119 | **0.252 (7)** |
| | std | 0.002 | 0.002 | 0.006 | 0.004 | 0.008 | 0.004 | 0.002 | - | 0.001 | 0.007 |
| | MRR | 0.130 | 0.157 | 0.161 | 0.09 | 0.044 | 0.068 | 0.085 | o.o.m | 0.000 | **0.166 (28)** |
| | std | 0.010 | 0.011 | 0.009 | 0.015 | 0.007 | 0.009 | 0.005 | - | 0.000 | 0.012 |

in Tables 4, 6, and 7, we successfully transfer HGNN models between author and venue nodes (A-V and V-A) for both L1 and L2 tasks.

Will lengths of meta-paths affect the performance? We examine the performance of KTN varying the length of meta-paths between source and target node types. In Table 8, accuracy decreases with longer meta-paths. When we add additional meta-paths than the minimum path, it also brings noise in every edge types. Note that author and venue nodes are indirectly connected by paper nodes; thus the minimum length of meta-paths in the A-V (L1) task is 2. The accuracy in the A-V (L1) task with a meta-path of length 1 is low because KTN fails to transfer anything with a meta-path shorter than the minimum. Using the minimum length of meta-paths is enough for KTN.

## A.3 More results for Zero-shot Transfer Learning in Section 6.3

We show the zero-shot transfer learning results with error bars on OAG-computer science and Pubmed in Tables 4 and 5. We also present the results with error bars on OAG-computer networks and OAG-machine learning in Tables 6 and 7, respectively. Across all tasks and graphs, our proposed method KTN consistently outperforms all baselines.

## A.4 Analysis for Baselines in Section 6.3

Among baselines, MMD-based models (DAN and JAN) outperform adversarial based methods (DANN, CDAN, and CDAN-E) and optimal transport-based method (WDGRL), unlike results reported in (19; 27). These reversed results are a consequence of HGNN's unique feature extractors for source and target domains. When $f_s$ and $f_t$ denote feature extractors for source and target domains, respectively, DANN and CDAN define their adversarial losses as a cross entropy loss ($\mathbb{E}[\log f_s] - \mathbb{E}[\log f_t]$) where gradients of the subloss $\mathbb{E}[\log f_s]$ are passed only back to $f_s$, while gradients of the subloss $\mathbb{E}[\log f_t]$ are passed only back to $f_t$. Importantly, source and target feature extractors do not share any gradient information, resulting in divergence. This did not occur in their original test environments where source and target domains share a single feature extractor. Similarly, WDGRL measures the first-order Wasserstein distance as an adversarial loss, which also brings the same effect as the cross-entropy loss we described above, leading to divergent gradients between source and target feature extractors. On the other hand, DAN and JAN define a loss in terms of higher-order MMD between source and target features. Then the gradients of the loss passed to

Table 5: **PubMed**

| Task | Metric | Base | DAN | JAN | DANN | CDAN | CDAN-E | WDGRL | LP | EP | KTN (gain%) |
|---|---|---|---|---|---|---|---|---|---|---|---|
| **D-G** | NDCG | 0.587 | 0.629 | 0.615 | 0.614 | 0.624 | 0.646 | 0.604 | 0.601 | 0.571 | **0.700 (19)** |
| | std | 0.004 | 0.013 | 0.028 | 0.008 | 0.078 | 0.015 | 0.022 | 0.000 | 0.004 | 0.005 |
| | MRR | 0.372 | 0.425 | 0.414 | 0.397 | 0.428 | 0.443 | 0.388 | 0.389 | 0.336 | **0.499 (34)** |
| | std | 0.003 | 0.007 | 0.054 | 0.013 | 0.066 | 0.027 | 0.035 | 0.000 | 0.003 | 0.006 |
| **G-D** | NDCG | 0.596 | 0.599 | 0.577 | 0.599 | 0.581 | 0.606 | 0.578 | 0.576 | 0.580 | **0.662 (11)** |
| | std | 0.007 | 0.020 | 0.032 | 0.011 | 0.054 | 0.019 | 0.019 | 0.000 | 0.011 | 0.003 |
| | MRR | 0.354 | 0.362 | 0.332 | 0.356 | 0.337 | 0.362 | 0.340 | 0.351 | 0.353 | **0.445 (26)** |
| | std | 0.005 | 0.015 | 0.019 | 0.008 | 0.023 | 0.031 | 0.015 | 0.000 | 0.008 | 0.002 |

Table 6: **Open Academic Graph on Computer Network field**

| Task | Metric | Base | DAN | JAN | DANN | CDAN | CDAN-E | WDGRL | LP | EP | KTN (gain%) |
|---|---|---|---|---|---|---|---|---|---|---|---|
| **P-A (L2)** | NDCG | 0.331 | 0.344 | o.o.m | 0.335 | o.o.m | o.o.m | 0.287 | 0.221 | 0.270 | **0.382 (16)** |
| | std | 0.004 | 0.005 | - | 0.004 | - | - | 0.012 | 0.000 | 0.003 | 0.004 |
| | MRR | 0.250 | 0.277 | o.o.m | 0.280 | o.o.m | o.o.m | 0.199 | 0.130 | 0.270 | **0.360 (44)** |
| | std | 0.024 | 0.012 | - | 0.007 | - | - | 0.004 | 0.000 | 0.003 | 0.010 |
| **A-P (L2)** | NDCG | 0.313 | 0.290 | o.o.m | 0.250 | 0.234 | 0.168 | 0.266 | 0.114 | 0.319 | **0.364 (17)** |
| | std | 0.002 | 0.023 | - | 0.021 | 0.041 | 0.025 | 0.030 | 0.000 | 0.004 | 0.003 |
| | MRR | 0.250 | 0.233 | o.o.m | 0.130 | 0.116 | 0.051 | 0.212 | 0.038 | 0.296 | **0.368 (47)** |
| | std | 0.015 | 0.039 | - | 0.051 | 0.069 | 0.037 | 0.061 | 0.000 | 0.005 | 0.004 |
| **A-V (L2)** | NDCG | 0.539 | 0.521 | 0.519 | 0.510 | 0.467 | 0.362 | 0.471 | 0.232 | 0.443 | **0.567 (5)** |
| | std | 0.012 | 0.031 | 0.008 | 0.022 | 0.008 | 0.045 | 0.024 | 0.000 | 0.002 | 0.008 |
| | MRR | 0.584 | 0.528 | 0.461 | 0.510 | 0.293 | 0.294 | 0.365 | 0.000 | 0.406 | **0.628 (8)** |
| | std | 0.042 | 0.015 | 0.011 | 0.054 | 0.013 | 0.088 | 0.019 | 0.000 | 0.004 | 0.016 |
| **V-A (L2)** | NDCG | 0.256 | 0.343 | 0.345 | 0.265 | 0.328 | 0.316 | 0.263 | 0.133 | 0.119 | **0.341 (33)** |
| | std | 0.006 | 0.012 | 0.005 | 0.005 | 0.005 | 0.003 | 0.003 | 0.000 | 0.001 | 0.005 |
| | MRR | 0.117 | 0.296 | 0.286 | 0.151 | 0.285 | 0.275 | 0.147 | 0.000 | 0.000 | **0.281 (141)** |
| | std | 0.020 | 0.009 | 0.004 | 0.009 | 0.006 | 0.008 | 0.009 | 0.000 | 0.000 | 0.014 |

each feature extractor contain both source and target feature information, resulting in a more stable gradient estimation. This shows again the importance of considering different feature extractors in HGNNs.

JAN, CDAN, and CDAN-E often show out of memory issues in Tables 4, 6, and 7. These baselines consider the classifier prediction whose dimension is equal to the number of classes in a given task. That is why JAN, CDAN, and CDAN-E fail at the L2 field prediction tasks in OAG graphs where the number of classes is $17,729$.

LP performs worst among the baselines, showing the limitation of relying only on graph structures. LP maintains a label vector with the length equal to the number of classes for each node, thus shows out-of-memory issues on tasks with large number of classes on large-size graphs (L2 tasks with $17,729$ labels on the OAG-CS graph). EP performs moderately well similar to other DA methods, but lower than KTN up to $60\%$ absolute points of MRR, showing the limitation of not using target node attributes.

## A.5 More results for Generality of KTN in Section 6.4

We show KTN performance on 6 different types of HGNN models across 4 different zero-shot domain adaptation tasks on the OAG-computer science dataset in Table 9. Descriptions of each HGNN model can be found in Appendix A.10. While KTN consistently improves all HGNN models' performance on zero-labeled node types using labels rooted at other node types, the magnitude of improvements varies. While HAN sees up to $4958\%$ (V-A (L1) task in Table 9), MAGNN is improved by up to $47\%$ (P-A(L1) task) or sees no improvement (A-V(L1) task). This gap stems from how many parameters each HGNN model shares across node types. HAN does not share any parameters during message-passing operations (every parameters are specialized to each meta-path), while MAGNN shares the transformation matrices across all node types at every layer. By sharing more parameters with other node types, the gradients are more likely passed to target node type-specific parameters, leaving less room for improvement by KTN. However, KTN is still necessary for any HGNN models. MPNN who shares all parameters except the projection matrices that map different input attributes into the same embedding space at the beginning still sees improvements by up to $311\%$. Again, these experimental results show the impact of having different feature extractors for each node type in HGNN models.

Table 7: **Open Academic Graph on Machine Learning field**

| Task | Metric | Base | DAN | JAN | DANN | CDAN | CDAN-E | WDGRL | LP | EP | KTN (gain%) |
|------|--------|------|-----|-----|------|------|--------|-------|----|----|-------------|
| **P-A (L2)** | **NDCG** | 0.268 | 0.290 | o.o.m | 0.291 | o.o.m | 0.249 | 0.232 | 0.272 | 0.215 | **0.318 (19)** |
| | **std** | 0.002 | 0.009 | - | 0.004 | - | 0.005 | 0.004 | 0.000 | 0.002 | 0.004 |
| | **MRR** | 0.134 | 0.220 | o.o.m | 0.222 | o.o.m | 0.095 | 0.098 | 0.195 | 0.143 | **0.269 (102)** |
| | **std** | 0.006 | 0.020 | - | 0.026 | - | 0.003 | 0.037 | 0.000 | 0.003 | 0.006 |
| **A-P (L2)** | **NDCG** | 0.261 | 0.225 | o.o.m | 0.234 | 0.228 | 0.241 | 0.241 | 0.119 | 0.267 | **0.319 (22)** |
| | **std** | 0.002 | 0.009 | - | 0.004 | 0.005 | 0.011 | 0.002 | 0.000 | 0.001 | 0.005 |
| | **MRR** | 0.207 | 0.127 | o.o.m | 0.155 | 0.152 | 0.095 | 0.182 | 0.035 | 0.214 | **0.287 (39)** |
| | **std** | 0.018 | 0.042 | - | 0.008 | 0.009 | 0.003 | 0.017 | 0.000 | 0.012 | 0.011 |
| **A-V (L2)** | **NDCG** | 0.465 | 0.493 | 0.463 | 0.477 | 0.408 | 0.422 | 0.393 | 0.224 | 0.424 | **0.538 (16)** |
| | **std** | 0.006 | 0.004 | 0.003 | 0.003 | 0.006 | 0.013 | 0.005 | 0.000 | 0.005 | 0.004 |
| | **MRR** | 0.469 | 0.542 | 0.537 | 0.519 | 0.412 | 0.240 | 0.213 | 0.001 | 0.391 | **0.632 (35)** |
| | **std** | 0.039 | 0.008 | 0.005 | 0.003 | 0.015 | 0.008 | 0.009 | 0.000 | 0.021 | 0.006 |
| **V-A (L2)** | **NDCG** | 0.252 | 0.293 | 0.292 | 0.237 | 0.242 | 0.255 | 0.250 | 0.137 | 0.119 | **0.302 (20)** |
| | **std** | 0.006 | 0.011 | 0.009 | 0.004 | 0.003 | 0.002 | 0.004 | 0.000 | 0.003 | 0.007 |
| | **MRR** | 0.131 | 0.212 | 0.199 | 0.086 | 0.085 | 0.129 | 0.118 | 0.000 | 0.000 | **0.227 (73)** |
| | **std** | 0.016 | 0.023 | 0.013 | 0.005 | 0.021 | 0.007 | 0.012 | 0.000 | 0.000 | 0.015 |

Table 8: **Meta-path length in KTN:** increasing the meta-path longer than the minimum does not bring significant improvement to KTN. Note that the minimum length of meta-paths in the A-V (L1) task is 2.

| Task | P-A (L1) | | A-V (L1) | |
|------|----------|------|----------|------|
| **Meta-path length** | **NDCG** | **MRR** | **NDCG** | **MRR** |
| **1** | 0.623 | 0.621 | 0.208 | 0.010 |
| **2** | 0.627 | 0.628 | 0.673 | 0.696 |
| **3** | 0.608 | 0.611 | 0.627 | 0.648 |
| **4** | 0.61 | 0.623 | 0.653 | 0.671 |

## A.6 Effect of trade-off coefficient $\lambda$

We examine the effect of $\lambda$ on transfer learning performance. In Table 10, as $\lambda$ decreases, target accuracy decreases as expected. Source accuracy also sees small drops since $\mathcal{L}_{\text{KTN}}$ functions as a regularizer; by removing the regularization effect, source accuracy decreases. When $\lambda$ becomes large, both source and target accuracy drop significantly. Source accuracy drops since the effect of $\mathcal{L}_{\text{KTN}}$ becomes larger than the classification loss $\mathcal{L}_{\text{CL}}$. Even the effect of transfer learning become larger by having larger $\lambda$, since the source accuracy which will be transferred to the target domain is low, the target accuracy is also low. Thus we set $\lambda$ to 1 throughout the experiments.

## A.7 Synthetic Heterogeneous Graph Generator

Our synthetic heterogeneous graph generator is based on attributed Stochastic Block Models (SBM) (32; 33), using blocks (clusters) as the node classes. In the attributed SBM, graphs exhibit *within-type* cluster homophily at the *edge-level* (nodes most-frequently connect to other nodes in the same cluster), and at the *feature-level* (nodes are closest in feature space to other nodes in the same cluster). To produce heterogeneous graphs, we additionally introduce *between-type* cluster homophily, which allows us to model real-world heterogeneous graphs in which knowledge can be shared across node types.

The first step in generating a heterogeneous SBM is to decide how many clusters will partition each node type. Assume *within-type* cluster counts $k_1, \ldots, k_T$. We allow for *between-type* cluster homophily with a $K_T := \min_t\{k_t\}$-partition of clusters such that each cluster group has at least one corresponding cluster from other node types.

Secondly, edge-level homophily is controlled by signal-to-noise ratios $\sigma_e = p/q$ where nodes *within-cluster* are connected with probability $p$ and nodes *between-cluster* are connected with probability $q$. Additionally, edges *within one cluster group across different types* (see previous paragraph) is controlled together with edges *between different cluster groups across different types* using some $\sigma_e$. In Section 6.5, we describe the manipulation of multiple $\sigma_e$ parameters *within-and-between* types.

Finally, node attributes are generated by a multivariate Normal mixture model, using the cluster partition as the mixture groups. Thus feature-level homophily is controlled by increasing the variance of the cluster centers $\sigma_f$, while keeping the within-cluster variance fixed. Cross-type feature

Table 9: **KTN on different HGNN models**: The *Source* column shows accuracy on source node types. *Base* and KTN columns show accuracy on target node types without/with using KTN, respectively. The *Gain* column shows the relative gain of our method over using no transfer learning.

| | | P-A (L1) | | | | A-P (L1) | | | |
|---|---|---|---|---|---|---|---|---|---|
| **HGNN type** | **Metric** | **Source** | **Base** | **KTN** | **Gain%** | **Source** | **Base** | **KTN** | **Gain%** |
| **R-GCN** | **NDCG** | 0.765 | 0.337 | 0.577 | **71.12** | 0.648 | 0.388 | 0.647 | **66.82** |
| | std | 0.004 | 0.005 | 0.002 | | 0.006 | 0.007 | 0.004 | |
| | **MRR** | 0.757 | 0.236 | 0.587 | **148.73** | 0.623 | 0.270 | 0.611 | **126.18** |
| | std | 0.002 | 0.003 | 0.001 | | 0.005 | 0.008 | 0.004 | |
| **HAN** | **NDCG** | 0.476 | 0.179 | 0.520 | **190.56** | 0.515 | 0.182 | 0.512 | **181.33** |
| | std | 0.004 | 0.006 | 0.003 | | 0.004 | 0.009 | 0.011 | |
| | **MRR** | 0.416 | 0.047 | 0.497 | **960.55** | 0.509 | 0.055 | 0.527 | **850.90** |
| | std | 0.001 | 0.002 | 0.002 | | 0.005 | 0.004 | 0.005 | |
| **HGT** | **NDCG** | 0.757 | 0.294 | 0.574 | **95.07** | 0.670 | 0.283 | 0.581 | **104.83** |
| | std | 0.002 | 0.003 | 0.004 | | 0.001 | 0.003 | 0.009 | |
| | **MRR** | 0.749 | 0.178 | 0.563 | **216.17** | 0.670 | 0.149 | 0.565 | **279.52** |
| | std | 0.005 | 0.007 | 0.001 | | 0.002 | 0.007 | 0.006 | |
| **MAGNN** | **NDCG** | 0.657 | 0.463 | 0.574 | **24.01** | 0.676 | 0.557 | 0.622 | **11.68** |
| | std | 0.003 | 0.001 | 0.003 | | 0.001 | 0.001 | 0.003 | |
| | **MRR** | 0.631 | 0.378 | 0.556 | **47.33** | 0.680 | 0.509 | 0.592 | **16.14** |
| | std | 0.003 | 0.002 | 0.004 | | 0.001 | 0.002 | 0.005 | |
| **MPNN** | **NDCG** | 0.602 | 0.443 | 0.590 | **33.11** | 0.646 | 0.307 | 0.621 | **101.92** |
| | std | 0.002 | 0.002 | 0.001 | | 0.005 | 0.013 | 0.004 | |
| | **MRR** | 0.572 | 0.319 | 0.575 | **80.10** | 0.660 | 0.145 | 0.595 | **311.42** |
| | std | 0.001 | 0.003 | 0.005 | | 0.002 | 0.024 | 0.003 | |
| **H-MPNN** | **NDCG** | 0.789 | 0.399 | 0.623 | **56.14** | 0.671 | 0.401 | 0.733 | **82.88** |
| | std | 0.001 | 0.005 | 0.001 | | 0.003 | 0.005 | 0.009 | |
| | **MRR** | 0.777 | 0.297 | 0.629 | **111.86** | 0.661 | 0.318 | 0.711 | **123.30** |
| | std | 0.003 | 0.001 | 0.002 | | 0.007 | 0.004 | 0.008 | |

| | | V-A (L1) | | | | A-V (L1) | | | |
|---|---|---|---|---|---|---|---|---|---|
| **HGNN type** | **Metric** | **Source** | **Base** | **KTN** | **Gain%** | **Source** | **Base** | **KTN** | **Gain%** |
| **R-GCN** | **NDCG** | 0.664 | 0.426 | 0.530 | **24.36** | 0.660 | 0.599 | 0.744 | **24.26** |
| | std | 0.003 | 0.006 | 0.002 | | 0.001 | 0.008 | 0.004 | |
| | **MRR** | 0.683 | 0.325 | 0.514 | **58.39** | 0.656 | 0.524 | 0.785 | **49.87** |
| | std | 0.003 | 0.008 | 0.004 | | 0.011 | 0.009 | 0.005 | |
| **HAN** | **NDCG** | 0.618 | 0.153 | 0.510 | **232.35** | 0.515 | 0.546 | 0.689 | **26.21** |
| | std | 0.005 | 0.007 | 0.003 | | 0.008 | 0.003 | 0.005 | |
| | **MRR** | 0.634 | 0.010 | 0.516 | **4958.82** | 0.508 | 0.511 | 0.758 | **48.28** |
| | std | 0.002 | 0.005 | 0.002 | | 0.001 | 0.008 | 0.007 | |
| **HGT** | **NDCG** | 0.615 | 0.234 | 0.536 | **128.95** | 0.694 | 0.367 | 0.735 | **100.22** |
| | std | 0.002 | 0.005 | 0.002 | | 0.006 | 0.007 | 0.009 | |
| | **MRR** | 0.638 | 0.095 | 0.537 | **464.88** | 0.699 | 0.267 | 0.772 | **189.21** |
| | std | 0.006 | 0.002 | 0.005 | | 0.002 | 0.005 | 0.012 | |
| **MAGNN** | **NDCG** | 0.536 | 0.513 | 0.513 | **0.00** | 0.684 | 0.676 | 0.692 | **2.37** |
| | std | 0.005 | 0.001 | 0.001 | | 0.001 | 0.002 | 0.001 | |
| | **MRR** | 0.586 | 0.522 | 0.522 | **0.00** | 0.686 | 0.751 | 0.752 | **0.13** |
| | std | 0.004 | 0.001 | 0.002 | | 0.002 | 0.001 | 0.004 | |
| **MPNN** | **NDCG** | 0.578 | 0.380 | 0.532 | **40.03** | 0.639 | 0.578 | 0.794 | **37.19** |
| | std | 0.008 | 0.008 | 0.004 | | 0.007 | 0.007 | 0.005 | |
| | **MRR** | 0.603 | 0.253 | 0.505 | **100.12** | 0.652 | 0.584 | 0.847 | **44.96** |
| | std | 0.001 | 0.003 | 0.007 | | 0.006 | 0.001 | 0.006 | |
| **H-MPNN** | **NDCG** | 0.670 | 0.283 | 0.584 | **106.50** | 0.676 | 0.459 | 0.671 | **46.22** |
| | std | 0.002 | 0.002 | 0.006 | | 0.005 | 0.004 | 0.003 | |
| | **MRR** | 0.689 | 0.133 | 0.586 | **339.76** | 0.677 | 0.364 | 0.698 | **91.92** |
| | std | 0.003 | 0.003 | 0.005 | | 0.01 | 0.005 | 0.002 | |

Table 10: **Effect of** $\lambda$

| | P-A (L1) | | | | A-V (L1) | | | |
|---|---|---|---|---|---|---|---|---|
| **Metric** | **NDCG** | | **MRR** | | **NDCG** | | **MRR** | |
| $\lambda$ | **Source** | **Target** | **Source** | **Target** | **Source** | **Target** | **Source** | **Target** |
| $10^{-5}$ | 0.780 | 0.587 | 0.772 | 0.595 | 0.689 | 0.626 | 0.690 | 0.642 |
| $10^{-3}$ | 0.788 | 0.58 | 0.779 | 0.576 | 0.687 | 0.654 | 0.689 | 0.677 |
| $10^{0}$ | 0.792 | 0.621 | 0.788 | 0.633 | 0.689 | 0.670 | 0.692 | 0.696 |
| $10^{2}$ | 0.750 | 0.617 | 0.757 | 0.623 | 0.654 | 0.644 | 0.659 | 0.668 |
| $10^{4}$ | 0.143 | 0.177 | 0.007 | 0.031 | 0.411 | 0.432 | 0.373 | 0.421 |

Table 11: **Statistics of Open Academic Graph**

| Domain | #papers | #authors | #fields | #venues | #institues | |
|---|---|---|---|---|---|---|
| **Computer Science** | 544,244 | 510,189 | 45,717 | 6,934 | 9,097 | |
| **Computer Network** | 75,015 | 82,724 | 12,014 | 2,115 | 4,193 | |
| **Machine Learning** | 90,012 | 109,423 | 19,028 | 3,226 | 5,455 | |
| **Domain** | **#P-A** | **#P-F** | **#P-V** | **#A-I** | **#P-P** | **#F-F** |
| **Computer Science** | 1,091,560 | 3,709,711 | 544,245 | 612,873 | 11,592,709 | 525,053 |
| **Computer Network** | 155,147 | 562,144 | 75,016 | 111,180 | 1,154,347 | 110,869 |
| **Machine Learning** | 166,119 | 585,339 | 90,013 | 156,440 | 1,209,443 | 163,837 |

Table 12: **Statistics of PubMed Graph**

| #gene | #disease | #chemicals | #species | |
|---|---|---|---|---|
| 13,561 | 20,163 | 26,522 | 2,863 | |
| **#G-G** | **#G-D** | **#D-D** | **#C-G** | **#C-D** |
| 32,211 | 25,963 | 68,219 | 31,278 | 51,324 |
| **#C-C** | **#C-S** | **#S-G** | **#S-D** | **#S-S** |
| 124,375 | 6,298 | 3,156 | 5,246 | 1,597 |

homophily is controlled by setting a center of cluster centers *within-type* with linear combinations of centers (of cluster centers) of other types. Note that features of different types are allowed to have different dimensions, as we generate different mixture-model cluster centers for each cluster *within each type*.

### A.7.1 Toy Heterogeneous Graph in Section 4.2

Using the synthetic graph procedure described above, we used the following hyperparameters to simulate the toy heterogeneous graph shown in Figure 2. We generate the graph with 2 node types and 4 edge types as described in Figure 1(a), then we divide each node type into 4 classes of 400 nodes. To generate an easy-to-transfer scenario, signal-to-noise ratio $\sigma_f$ between means of feature distributions are all set to 10. The ratio $\sigma_e$ of the number of intra-class edges to the number of inter-class edges is set to 10 among the same node types and across different node types. The dimension of features is set to 24 for both node types.

### A.7.2 Sensitivity test in Section 6.5

Figure 5(a) shows the structures of graphs we used in Section 6.5. The dimension of features are set to 24 for both node types for the "easy" scenario, and 32 and 48 for types $s$ and $t$, respectively, for the "hard" scenario. Additionally, for the "hard" scenario, we divide the $t$ nodes into 8 clusters instead of 4. The other hyperparameters $\sigma_e$ and $\sigma_f$ are described in Section 6.5. For each unique value of $\sigma_{(\cdot)}$ across the six $(\sigma_{(\cdot)}, r)$ pairs, we generate 5 heterogeneous graphs.

### A.8 Real-world Dataset

**Open Academic Graph (OAG)** (28; 31; 44) is the largest publicly available heterogeneous graph. It is composed of five types of nodes: papers, authors, institutions, venues, fields and their corresponding relationships. Papers and authors have text-based attributes, while institutions, venues, and fields have text- and graph structure-based attributes. To test the generalization of the proposed model, we construct three field-specific subgraphs from OAG: the Computer Science (OAG-CS), Computer Networks (OAG-CN), and Machine Learning (OAG-ML) academic graphs.

Papers, authors, and venues are labeled with research fields in two hierarchical levels, L1 and L2. OAG-CS has both L1 and L2 labels, while OAG-CN and OAG-ML have only L2 labels (their L1 labels are all "computer science"). Transfer learning is performed on the L1 and L2 field prediction tasks between papers, authors, and venues for each of the aforementioned subgraphs. Note that paper-author (P-A) and paper-venue (P-V) are directly connected, while author-venue (A-V) are indirectly connected via papers.

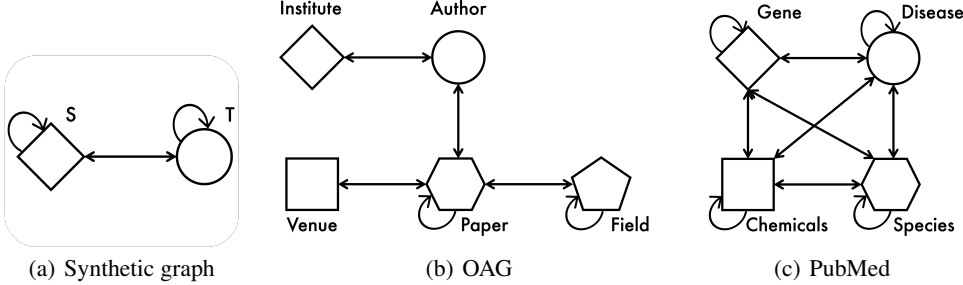

|(a) Synthetic graph | (b) OAG | (c) PubMed |

Figure 5: Schema of synthetic and real-world heterogeneous graphs

The number of classes in the L1 task is 275, while the number of classes in the L2 task is 17, 729. The graph statistics are listed in Table 11, in which P–A, P–F, P–V, A–I, P–P, and F-F denote the edges between paper and author, paper and field, paper and venue, author and institute, the citation links between two papers, and the hierarchical links between two fields. The graph structure is described in Figure 5(b).

For paper nodes, features are generated from each paper's title using a pre-trained XLNet (36). For author nodes, features are averaged over features of papers they published. Feature dimension of paper and author nodes is 769. For venue, institution, and field node types, features of dimension 400 are generated from their heterogeneous graph structures using metapath2vec (5).

**PubMed** (39) is a novel biomedical network constructed through text mining and manual processing on biomedical literature. PubMed is composed of genes, diseases, chemicals, and species. Each gene or disease is labeled with a set of diseases (e.g., cardiovascular disease) they belong to or cause. Transfer learning is performed on a disease prediction task between genes and disease node types.

The number of classes in the disease prediction task is 8. The graph statistics are listed in Table 12, in which G, D, C, and S denote genes, diseases, chemicals, and species node types. The graph structure is described in Figure 5(c).

For gene and chemical nodes, features of dimension 200 are generated from related PubMed papers using word2vec (23). For diseases and species nodes, features of dimension 50 are generated based on their graph structures using TransE (4).

## A.9 Baselines

Zero-shot domain adaptation can be categorized into three groups — MMD-based methods, adversarial methods, and optimal-transport-based methods. MMD-based methods (18; 20; 29) minimize the maximum mean discrepancy (MMD) (11) between the mean embeddings of two distributions in reproducing kernel Hilbert space. DAN (18) enhances the feature transferability by minimizing multi-kernel MMD in several task-specific layers. JAN (20) aligns the joint distributions of multiple domain-specific layers based on a joint maximum mean discrepancy (JMMD) criterion.

Adversarial methods (9; 19) are motivated by theory in (2; 3) suggesting that a good cross-domain representation contains no discriminative information about the origin of the input. They learn domain invariant features by a min-max game between the domain classifier and the feature extractor. DANN (9) learns domain invariant features by a min-max game between the domain classifier and the feature extractor. CDAN (19) exploits discriminative information conveyed in the classifier predictions to assist adversarial adaptation. CDAN-E (19) extends CDAN to condition the domain discriminator on the uncertainty of classifier predictions, prioritizing the discriminator on easy-to-transfer examples.

Optimal transport-based methods (27) estimate the empirical Wasserstein distance (25) between two domains and minimizes the distance in an adversarial manner. Optimal transport-based method are based on a theoretical analysis (25) that Wasserstein distance can guarantee generalization for domain adaptation. WDGRL (27) estimates the empirical Wasserstein distance between two domains and minimizes the distance in an adversarial manner.

## A.10 HGNN models

We briefly describe 6 heterogeneous graph neural networks (HGNN) models we used in the experiments. MPNN (message passing neural networks) (10) is originally designed for homogeneous graphs. We extend MPNN to process heterogeneous graphs by adding projection matrices that project input attributes of different node types into the same feature space before running the original MPNN. R-GCN (26) extends MPNN by specializing message matrices in each edge type, while HMPNN specializes all transformation and message matrices in each node/edge type in MPNN. HGT (15) extends HMPNN by adding attention modules. The attention modules have node-type-specific key/query projection matrices and edge-type-specific key-query similarity matrices, following the transformer architecture.

HAN (35) is a meta-path-based model who specializes parameters in each meta-path. HAN exploits meta-path-specific attention modules to aggregate features of neighboring nodes connected by each meta-path. Then HAN aggregates embeddings of different meta-paths with semantic-level attention modules. MAGNN (8) is another meta-path-based HGNN model. MAGNN aggregates features of all nearby nodes sitting on each meta-path using intra-meta-path attention modules. Then MAGNN aggregates features of different meta-paths using inter-meta-path attention modules.

All HGNN models we describe above have layer-wise parameters. As all HGNN models have parameters specialized in either node/edge/meta-path types, they all have distinct feature extractors for each node types, thus, they will suffer from the under-trained target node phenomena we showed in Section 4.2. Also, because the core intuition in KTN — namely that embeddings of any node types at the last layer are computed using the same set of the previous layer's intermediate embeddings (see Section 4.3) — holds across all HGNN models, KTN can be applied to any HGNN models and show greatly increased target-type accuracy.

## A.11 Experimental Settings

All experiments were conducted on the same p2.xlarge Amazon EC2 instance. Here, we describe the structure of HGNNs used in each heterogeneous graph.

**Open Academic Graph:** We use a 4-layered HGNN with transformation and message parameters of dimension 128 for KTN and other baselines. Learning rate is set to $10^{-4}$.

**PubMed:** We use a single-layered HGNN with transformation and message parameters of dimension 10 for KTN and other baselines. Learning rate is set to $5 \times 10^{-5}$.

**Synthetic Heterogeneous Graphs:** We use a 2-layered HGNN with transformation and message parameters of dimension 128 for KTN and other baselines. Learning rate is set to $10^{-4}$.

We implement LP, EP and KTN using Pytorch. For the domain adaptation baselines (DAN, JAN, DANN, CDAN, CDAN-E, and WDGRL), we use a public domain adaptation library ADA [3]. We match the numbers of layers and dimensions of hidden embeddings across all HGNN models. We implement MPNN and HMPNN using Pytorch. For other HGNN models (R-GCN, HAN, HGT, and MAGNN), we use an open-source toolkit for Heterogeneous Graph Neural Network (OpenHGNN) [4]. Our code is publicly available [5].

---

[3] https://github.com/criteo-research/pytorch-ada
[4] https://github.com/BUPT-GAMMA/OpenHGNN
[5] https://github.com/minjiyoon/KTN