# OpenReview forum: "Zero-shot Transfer Learning within a Heterogeneous Graph via Knowledge Transfer Networks"
_NeurIPS.cc/2022/Conference — NeurIPS 2022 Accept_

### Official Review · Reviewer_AyUC · 2022-07-04

**Rating:** 7
**Confidence:** 4
**Soundness:** 4 excellent
**Presentation:** 4 excellent
**Contribution:** 3 good

**Summary:**

This paper aims to transfer knowledge from the source node type with abundant labels to the target node type with zero labels on heterogeneous graphs by proposing a knowledge transfer network. In particular,
* The authors introduce the undiscovered problem of transferring knowledge from label-abundant nodes to zero-labeled nodes, which are connected via edges over the heterogeneous graph.
* The authors tackle this problem with a novel knowledge transfer network, which trains a feature transformation matrix that maps representations for target nodes with zero-label to the source node representation space.
* The authors propose a knowledge transfer network with a motivating theoretical observation that there exist transformation matrices to exactly map the target node representations to the source node embedding space, for predicting target zero-labeled nodes over the learned source-node representation space.
* The experimental results show that the proposed knowledge transfer network significantly outperforms the no domain adaptation and existing domain adaptation methods, by a large margin.

**Questions:**

### Major Questions and Suggestions
* The authors may discuss the relevant work on transfer learning or domain adaptation on graphs with graph neural networks (See the first weakness above).
* The authors should improve the explanations on synthetic graph experiments (See the second weakness above).
* The authors do not explain the task (e.g., P-A (L1)) in Table 1 and Table 2. I guess P-A denotes the Paper (Source) and Author (Target) as node types, however, the authors should define this.
* It would be interesting to see the performance on few-shot tasks, where few target nodes have their labels.

### Minor Questions and Suggestions
* The description of the first contribution in Line 76 is a little bit unclear. It is better to specify the term "cross-type transfer learning".
* Does the number of batchs in Figure 2 denote the number of iterations, or the batch size? It seems it would be the number of iterations, but the current term of the number of batchs is ambiguous.
* I am just curious that, instead of minimizing the loss for KTN for the representation of the last layer (i.e., L), why the authors do not minimize the KTN losses for all layers (i.e., 1, 2, 3, ..., L).
* A caption for Table 2 is not informative.

### Typo
* Line 88: in in Section 3.

**Limitations:**

The authors provide the limitations and potential societal impact of their work. While the authors do not describe the *negative* societal impact, but only describe the positive point, this does not hurt the contributions of this work.

**Strengths And Weaknesses:**

### Strengths
* This paper tackles the highly realistic problem of zero-shot knowledge transfer from source to target nodes on a heterogeneous graph, which has, to my knowledge, not been introduced so far.
* The proposed method of representing zero-labeled target nodes in terms of label-abundant source nodes with the transformation matrices from target to source is convincing and highly interesting. I enjoy reading this method.
* The proposed architecture, namely knowledge transfer networks, is well-motivated by empirical observations to a toy heterogeneous graph, and by theoretical derivations to feature extraction modules on heterogeneous graph neural networks.
* The performance improvements from baselines are significant and impressive.
* This paper is extremely well written and easy to follow.

### Weaknesses
Note that the below points are not the major limitations of this work, but I would appreciate if the authors address them.
* The authors may include the work [1, 2] that considers transfer learning (or domain adaptation) problems for graph-structured data with graph neural networks, and perhaps more work on this topic of graph transfer learning could be discussed.
* The explanations of synthetic graph experiments in Section 4.2 and Section 6 are unclear. Also, the explanation in Appendix A.7, which contains full descriptions, is confusing as well. This might be because the setting for heterogeneous graphs differs from normal graphs, and I think it would be better to illustrate the authors' synthetic graph experimental setup with Figure.

---

[1] You et al. Handling Missing Data with Graph Representation Learning. NeurIPS 2020.

[2] Wu et al. Towards Open-World Feature Extrapolation: An Inductive Graph Learning Approach. NeurIPS 2021.

---

> ### Author Response · Authors · 2022-08-02
> **Response to R4**
>
> Thank you for the comments and suggestions!
>
> **Q1. More work on this topic of graph transfer learning could be discussed**
>
> Thank you for the references. We describe more works on graph transfer learning in Section 2, including two papers you recommended. There are many transfer learning methods running on graphs. An array of works, including those two papers [1, 2], do not handle graph-structured data but build synthetic graphs temporarily to solve the transfer learning problem efficiently. For instance, [1] constructs a bipartite graph where each observation and feature is viewed as two types of nodes and runs GNNs over the graph to impute features and predict labels. Another array of works focuses on extracting knowledge from given graph-structured data. Most of them assume to have an external source graph to extract knowledge from and another target graph to infuse the knowledge. Thus, they cannot be applied to our problem setting where no other external source graph is given.
>
> [1] NeurIPS 2020, Handling Missing Data with Graph Representation Learning
> [2] NeurIPS 2021, Towards Open-World Feature Extrapolation: An Inductive Graph Learning Approach
>
> **Q2. The explanations of synthetic graph experiments are unclear; recommend illustrating the authors' synthetic graph experimental setup with figures.**
>
> Following your suggestions, we add Figure 3 and an additional description to Section 6.5 to help explain how the generator manipulates feature and edge distributions of synthetic heterogeneous graphs.
>
> **Q3. The authors do not explain the task (e.g., P-A (L1)) in Table 1 and Table 2. I guess P-A denotes the Paper (Source) and Author (Target) as node types; however, the authors should define this**
>
> Previously, we explained them in Appendix A.8. Real-world Dataset. According to your suggestion, we describe the meaning of abbreviations in Section 6.1 before we use them. For your convenience, we describe the notation briefly here: In the OAG graphs, P, A, I, V, and F denote paper, author, institution, venue, and field node types. For instance, P-A (L1) denotes transfer learning from paper node types to author node types on the L1 research field prediction task. In the Pubmed graph, G, D, C, and S denote gene, disease, chemical, and species node types. D-G denotes transfer learning from disease node types to gene node types on the disease category (they belong to or cause) prediction task.
>
> **Q4. Few-shot learning performance**
>
> We run initial experiments on few-shot learning performance. As we can guess, the task becomes easier with more labels for the target nodes; thus, the performance gap between a base model with no transfer learning module and the model with KTN becomes decreasing (i.e., fewer benefits from KTN with more target labels). We leave out how to reap maximum benefits from a few target labels using KTN for our future research.
>
> **Q5.The description of the first contribution in Line 76 is a little bit unclear. It is better to specify the term "cross-type transfer learning"**
>
> We change it to “a cross-type transfer learning method running on a heterogeneous graph — transfer knowledge across different node types within a heterogeneous graph”. To clarify the term "cross-type transfer learning" even further, we have added a "formal" version of the problem statement to Section 3.3, making use of the rigorous notation we introduced. We hope this will clarify the issue!
>
> **Q6. Does the number of batches in Figure 2 denote the number of iterations or the batch size?**
>
> We change the x-axis labels to “Number of iterations” in Figure 2.
>
> **Q7. Instead of minimizing the loss for KTN for the representation of the last layer (i.e., L), why do the authors not minimize the KTN losses for all layers (i.e., 1, 2, 3, ..., L)?**
>
> The focus of this paper is zero-shot transfer learning — pretrain an HGNN model and a classifier on the source domain, then re-use them on the target domain without using any target labels. Thus we focus on learning a mapping matrix from the target domain to the source domain to re-use a classifier pretrained on the source domain by projecting target embeddings into the source domain using the mapping matrix. In other words, we do not need to learn mapping matrices for all layers, but only for one layer for one-time mapping from the target domain to the source domain. The idea of matching every intermediate layer could be a reasonable direction for representation learning. And it is obviously feasible but not our focus, so we leave it for future work.
>
> **Q8. A caption for Table 2 is not informative**
>
> We add more captions describing columns.
>
> **Q9. Typo: Line 88 “in in” Section 3**
>
> Thank you for pointing it out. We correct it.

---

> > ### Comment · Reviewer_AyUC · 2022-08-07
> > **Thank you for your response, and I maintain my score: accept.**
> >
> > I sincerely appreciate the authors' response, which sufficiently addresses all my previous concerns/comments. I recapitulate my questions and the authors' answers one by one as follows:
> >
> > Q1. Thank you for discussing relevant transfer learning literature, which looks good and is convincing for me.
> >
> > Q2. Thank you for providing detailed experimental setups for synthetic graphs in Lines 326-331 with Figure 3, which now clearly describe the experimental setups.
> >
> > Q4. Regarding additional experiments on few-shot learning, I suggest the authors include the results and then discuss them in the next revision. The poor performances on few-shot tasks do not hurt the contribution of this work since this work targets zero-shot settings, however, it is worthwhile to see the performance on few-shot tasks as well.
> >
> > Q7. Thank you for explanaing why the authors aim to learn the mapping matrix from the target domain to the source domain for the last layer, instead of all layers. In my understanding, learning all-layer mapping matrices (i.e., $H_t^1$,  ..., $H_t^L$ to the source domain) is not easy for the zero-shot setting, thus the authors target one last layer (i.e., $H_t^L$ to the source domain). Is this correct? If that is correct, the authors' explanation in the response comment makes sense.
> >
> > Q3, Q5, Q6, Q8, and Q9. Thank you for clarifying/correcting my minor questions/concerns.

---

> > > ### Author Response · Authors · 2022-08-09
> > > **Thank you for the great comments**
> > >
> > > We really appreciate all your comments and questions, which have helped us to improve the quality of the paper.
> > >
> > > All your recapitulation is aligned with what we had tried to deliver.
> > > For *Q7. why not learn mapping matrices for all layers?*, we can supplement your understanding as follows:
> > >
> > > We didn't learn mapping matrices for all layers because learning a mapping matrix only for the last layer is already sufficient for zero-shot transfer learning. **We only need to map the target embeddings into the source embedding space a single time.**
> > >
> > > Learning mapping matrices for other layers is feasible by defining additional knowledge transfer losses. For instance, \\(\mathcal{L_{ktn}}^{(l)} = ||H_{s}^{(l)} - A_{ts} H_{t}^{(l)} T_{ts}^{(l)}||\\) for \\((l = 1, \cdots, L) \\). In this case, \\(T_{ts}^{(l)}\\) becomes a mapping matrix for the \\(l\\)-th layer. **However, it's unclear how to exploit $L$ different mapping matrices for transfer learning.**

---

> > > > ### Comment · Reviewer_AyUC · 2022-08-09
> > > > **Thank you so much for clarifying all my questions/comments**
> > > >
> > > > After having a discussion with the authors, all my questions/comments are clearly resolved. Thank you again for responding to my comments, and engaging with me to clarify my questions. I am willing to maintain my score: accept.

---

### Official Review · Reviewer_Rrs6 · 2022-07-07

**Rating:** 8
**Confidence:** 4
**Soundness:** 3 good
**Presentation:** 3 good
**Contribution:** 3 good

**Summary:**

This paper notes that many HG datasets suffer from label imbalance between node types, and proposes a zero-shot transefer learning module to transfer knowledge from label-abundant node types to zero-labeled node types. This is a good work in my opinion.

**Questions:**

Please respond to the weaknesses raised above.

**Ethics Review Area:**

["I don’t know"]

**Limitations:**

The authors have well discussed the limitations and potential negative social impact of this work.

**Strengths And Weaknesses:**

- Strengths
  - This work notes a good research problem. The issue of label imbalance among different node types is common in existing HG datasets.
  - The proposed method KTN has a solid theoretical foundation, and its implementation is quite easy.
  - The experimental results show that KTN is very effective in improving the performance of existing HGNN models.
  - The provided source codes and datasets in the supplemental material facilitates the good reproducibility of this work.

- Weaknesses
  - In section 3.2, the authors have reviewed some HGNN methods, but one recent HGNN method is missing, i.e., *[TKDE 2021] Interpretable and Efficient Heterogeneous Graph Convolutional Network*.
  - For the training step and the test step, why the formula in Line 2 of Algorithm 1 has the adjacency matrix $A_{ts}$ while the formula in Line 2 of Algorithm 2 does not?
  - Is HMPNN an existing work? If so, it should be cited. The authors state that HMPNN merely extends the standard MPNN, but why does the message function (Eq. 2) in this work look different from the message function (Eq. 1) in MPNN?

---

> ### Author Response · Authors · 2022-08-02
> **Response to R3**
>
> Thank you for acknowledging our work! Answers to your questions follow.
>
>
> **Q1. Add citation: Interpretable and Efficient Heterogeneous Graph Convolutional Network**
>
> Thanks for the reference. We have added ie-HGCN as an example of HGNN models in Section 3.2. For your information, ie-HGCN has parameters specialized for each node and edge type; thus, it will suffer the same issue as other HGNN models — target node type-specific parameters are un/under-trained during pretraining on the source domain. Our proposed KTN can be simply applied between ie-HGCN and any task-specific output layer and help to improve zero-shot learning on target node types.
>
>
> **Q2. For the training and test step, why the formula in Line 2 of Algorithm 1 has the adjacency matrix while the formula in Line 2 of Algorithm 2 does not?**
>
> Equation 8 in Theorem 1 indicates that once the target embeddings \\(H_{t}^{(L)}\\) are mapped to the source domain using the mapping matrix \\(Q_{ts}^{\ast}\\), the source embeddings \\(H_{s}^{(L)}\\) can be represented with the neighboring t-type nodes’ (mapped) embeddings (\\(H_t^{(L)}Q_{ts}^{\ast}\\)) using the connectivity information (\\(A_{ts}^{\ast}\\)). Based on Equation 8, during the training step, we optimize \\(T_{ts}\\) to match \\(Q_{ts}^{\ast}\\) by minimizing L2 distance between the source embeddings \\(H_s^{(L)}\\) with the average of connected (mapped) target embeddings (\\(A_{ts}H_t^{(L)}T_{ts}\\)). That is why we multiply with \\(A_{ts}\\) in *Algorithm 1. training step*. On the other hand, during the test phase, we do not need to aggregate target node embeddings to match source node embeddings because our interest is the target node embeddings (mapped into the source domain), which will be fed into the task-specific output layer to predict its labels. That is why we multiply \\(H_t^{(L)}\\) only with \\(T_{ts}\\) and pass it to the classifier in *Algorithm 2. test step*.
>
>
> **Q3. Is HMPNN an existing work? If so, it should be cited.**
>
> HMPNN is our acronym for "Heterogeneous Message-Passing Neural Network", which we use to denote a trivial extension of the well-known MPNN to heterogeneous graphs. The HMPNN has already been used extensively in software packages (e.g.,  HMPNN is a default heterogeneous graph model in popular GNN libraries such as DGL, PyG, TF-GNN), but to our knowledge has never been explicitly published as a stand-alone method due to its relative simplicity. Because the MPNN is a generalization of nearly all GNNs, we choose the HMPNN as our analysis and implementation starting point for KTN. However, our proposed method can be applied to any HGNN model, as shown in Table 3.
>
>
> **Q4. The authors state that HMPNN merely extends the standard MPNN, but why does the message function (Eq. 2) in this work look different from the message function (Eq. 1) in MPNN?**
>
> The only difference between Equation 2 in our paper and Equation 1 in the original MPNN paper is the existence of edge attributes \\(e_{uv}\\). MPNN is designed specifically for molecule graphs, and molecule graphs are provided with edge attributes as edge attributes play a crucial role in deciding the properties of the molecules. Thus MPNN considers how to deal with edge attributes in Equation 1. On the other hand, many social, citation, or e-commerce graphs are given without edge attributes. The reason we choose HMPNN for analyzing the problem is its generality and simplicity compared to other HGNN models. Thus we omitted edge attributes in Equation 2. However, our proposed KTN is agnostic to HGNN models as KTN deals with the final node embeddings computed from the HGNN models, not the HGNN architectures. In other words, we can easily apply KTN on HGNN models designed to deal with edge attributes on heterogeneous graphs given with edge attributes.

---

> > ### Comment · Reviewer_Rrs6 · 2022-08-05
> > **Thanks for the response**
> >
> > Thank the authors for responding to my concerns. I will keep my rating score of 8, which is quite high. This is a good paper, and I strongly recommend accepting this work.

---

> > > ### Author Response · Authors · 2022-08-09
> > > **Thanks for the review**
> > >
> > > We really appreciate your acknowledgment of our work and your intuitive comments that helped us to improve our paper.

---

### Official Review · Reviewer_deGV · 2022-07-11

**Rating:** 6
**Confidence:** 3
**Soundness:** 2 fair
**Presentation:** 3 good
**Contribution:** 3 good

**Summary:**

This paper studies the zero-shot node types in knowledge transfer and proposes a model to transfer knowledge from label-abundant node types to zero-labeled ones. The main contributions of this paper lie in the proposed practical scenario and problem definition, i.e., knowledge transfer of zero-shot node types. The transferability of node label knowledge is theoretically analyzed. A node type label knowledge transfer network is proposed.

**Questions:**

This work proposes a new practical problem definition, which is beneficial to the development of the field. But I have a few confusing questions about the paper. Please point out if I misunderstood.

- In the problem definition, the authors say that only source type labels are available. Although the target labels are not used as training samples, the training graph data seems to contain the labels of the target type (i. target type related aggregation and ii. output embeddings of the nodes of the target type). Is this reasonable? This is my biggest concern.

- Does the proposed method still work on other graph tasks (such as link prediction) and on more complex graphs (such as knowledge graphs)? No need to add experiments, just give a brief statement about feasibility.

- If needed, you can respond to the above *Weaknesses* to help me make a better assessment.

**Limitations:**

N/A.

**Strengths And Weaknesses:**

Strengths:

- This paper studies a novel and practical problem, and the motivation is clearly stated.

- The proposed graph neural network is supported by theoretical analysis, although some of the analysis is based on a simple toy graph.

- Experimental results show that the proposed method outperforms the baseline methods.

Weaknesses:

- The title is misleading. There is a lot of transferable knowledge in the graph data, but this paper only discusses a specific setting of the node type labels.

- The baseline methods are from 2005–2018. Some new work needs to be discussed.

- As a work that proposes a new problem setting, Section 6.3 needs to provide more details of the experimental setting.

---

> ### Author Response · Authors · 2022-08-02
> **Response to R2 (1/3)**
>
> Thank you for acknowledging the novelty of our work! We hereby carefully address your concerns as follows:
>
> **Q1. The training graph data seems to contain the labels of the target type (i. target type-related aggregation and ii. output embeddings of the nodes of the target type). Is this reasonable? This is my biggest concern**
>
> We would like to clarify that **label** is not equivalent (nor dependent) to the target **node type**, and there isn’t label leakage in our training procedure. To sum up:
> - We denote labels as to which class each node belongs to in our target downstream classification task.
>   - For example, in academic graphs, we consider two types of nodes, "paper" and "author". For each paper instance, the task is to predict which research field it belongs to (e.g., ML, CV, DB are the task labels).
> - Target node type is independent of the target node labels; there isn’t leakage of information from the labels to the type itself.
>   - For example, in the academic graphs, the “author” node type itself does not tell anything about which research field each “author” node belongs to.
> - We do not use the node labels as part of input attributes (neither the source nor the target node types). In other words, none of the labels are used as input to HGNN models.
> - Furthermore, we do not test our transfer learning performance on the nodes from the source domain i.e., we test on target-typed nodes that we have never seen labels.
>
> To expound on this in detail:
>
> During the training stage, we are given (1) graph information — how nodes are connected with each other, (2) (publicly available) input node attributes for all node types, and (3) labels of source type nodes. This input information is exactly the same as the standard semi-supervised learning on HGNN models requires. Our novel problem definition adds a challenging task to this standard-setting: labels of target type nodes are completely missing during the training stage; can we predict the labels of target type nodes only using information (1), (2), and (3)?
>
> We train two models, an HGNN model followed by a classifier in an end-to-end manner. The HGNN receives (1) graph information and (2) input node attributes and outputs node embeddings for all node types (please note that labels are not fed into the HGNN). Then the classifier receives node embeddings and predicts labels. During the training stage, we feed embeddings of source-typed nodes into the classifier and compute a classification loss using (3) source labels. During the test time, we feed embeddings of target-typed nodes (which are also computed by the same HGNN) into the classifier and predict target labels. *Please note that we never use target labels during the training stage.*
>
> All the target-related information you mentioned (i. target type-related aggregation and ii. output embeddings of the nodes of the target type) are computed only using (1) graph information and (2) input node attributes for all node types.
> Let us give you an example. In an e-commerce network, you have access to product information (input node attribute for product node types) and product categories (labels for product node types). You also have access to publicly available reviews written by users (input node attribute for user node types), but not to private user preferences (labels for user node types). In this case, we want to train an HGNN model only using (publicly available) input node attributes for product/user nodes and (publicly available) product labels, then predict (unseen) user labels.
>
> Sometimes, heterogeneous graphs do not provide input attributes for all node types. In this case,  we can compute attributes based on graph structures or attributes on other node types. For instance, OAG graphs that are widely used in HGNN experiments (including ours) do not have input attributes for author, venue, institution, and field node types. We can build input attributes by a) averaging the connected nodes’ input attributes (e.g., the initial attributes of each author is an average of his/her published papers’ attributes) or b) computing graph-structure-based attributes (e.g., venue, institution, and field node types are initialized with TransE or metapath2vec embeddings that reflect heterogeneous graph structures).
>
> To clarify our problem setting even further, we have added a "formal" version of the problem statement to Section 3.3, making use of the rigorous notation we introduced. We hope this will clarify the issue!

---

> > ### Author Response · Authors · 2022-08-02
> > **Response to R2 (2/3)**
> >
> > **Q2. Does the proposed method still work on other graph tasks (such as link prediction) and on more complex graphs (such as knowledge graphs)?**
> >
> > First of all, our method is “task-agnostic” since aligning source and target domains happens in the representation space. We map the target node embeddings (computed from the HGNN) into the source embedding space before feeding them into task-specific output layers. Then any task-specific output layer that is trained with source node labels can be re-used for target node types with our method.
> >
> > Secondly, Knowledge Graphs (KGs) are a special case of heterogeneous graphs with large numbers of node and edge types.  Since there are modifications for other HGNN models to deal with this high number of relationships [1, 2], KTN should extend easily into these domains.  This is a great area for future research. Another exciting research in this area might be to extend KG embedding methods that do not use GNNs (e.g. the TransE family [3]) with KTN.
> >
> > [1] Schlichtkrull, Michael, et al. "Modeling relational data with graph convolutional networks." European semantic web conference. Springer, Cham, 2018.
> > [2] Vashishth, Shikhar, et al. "Composition-based multi-relational graph convolutional networks." arXiv preprint arXiv:1911.03082 (2019).
> > [3] Bordes, Antoine, et al. "Translating embeddings for modeling multi-relational data." Advances in neural information processing systems 26 (2013).
> >
> > **Link Prediction**
> >
> > To answer your question, we ran a simple link prediction task on the OAG-computer science graph. We select all the authors with the same name and their associated papers. During training, we conduct link predictions between papers written by the same author. After getting node embeddings from an HGNN model, we use a Neural Tensor Network (NTN) to get the probability of each paper-paper pair being linked. Using the pretrained HGNN and NTN models, we predict paper-author links. Note that we remove the paper-author links we will predict at the test phase from the graph.
> >
> > | Metric | Source (paper-paper) | Target (paper-author) | KTN (paper-author) | Improvement |
> > |--------|--------------------------|---------------------------|------------------------|-------------|
> > | NDCG   |           0.793          |           0.565           |          0.629         |         11% |
> > | MRR    |           0.609          |           0.358           |          0.506         |         41% |
> >
> > The table shows that KTN improves the link prediction performance by up to 41% on zero-labeled paper-author links. This shows the generality of KTN. We will add all those results and analyses to the final manuscript.
> >
> > **Q3. The title is misleading; should be more specific**
> >
> > In our problem setting, knowledge transfer happens within a single heterogeneous graph — from a high-resource (well-labeled) node type to a low-resource (un-labeled) node type. More specifically, the knowledge we are transferring between node types is an HGNN model and a classifier pretrained on the source node type. Then the target node type re-uses the HGNN and classifier (without using any target labels to finetune the models) with our proposed Knowledge Transfer Networks (KTN). Therefore we refine our title to “Zero-shot Transfer Learning **within** a Heterogeneous Graph via Knowledge Transfer Networks”.

---

> > > ### Author Response · Authors · 2022-08-02
> > > **Response to R2 (3/3)**
> > >
> > > **Q4. Compare with more recent domain adaptation methods.**
> > >
> > > We run two more recent domain adaptation baselines, ALDA [1] (AAAI 2020) and ToAlign [2] (Neurips 2021), on the zero-shot transfer learning tasks on the OAG graph (the same experiments we described in Table 1 in Section 6.3).
> > > ALDA [1] combines domain-adversarial learning and self-training to reduce the gap and align the feature distributions. It learns a confusion matrix through an adversarial manner to correct the pseudo-label of the unlabeled target sample. ToAlign [2] makes the domain alignment proactively serve classification by aligning target features with source task-discriminative features, which are obtained under the guidance of meta-knowledge induced from the classification task. Both ALDA and ToAlign assume the source and target domains share the same feature extractors (as all other DA methods assume). This assumption results in poor DA performance on HGNNs with a unique feature extractor for each domain.
> > >
> > > |   Task   | Metric | Source |  ALDA | ToAlign | KTN (ours) | Improvement |
> > > |:--------:|:------:|:------:|:-----:|:-------:|:----------:|:-----------:|
> > > | P-A (L1) | NDCG   |  0.399 | 0.235 |  0.472  |    0.623   |         32% |
> > > |          | MRR    |  0.297 | 0.096 |  0.451  |    0.629   |         39% |
> > > | A-P (L1) | NDCG   |  0.401 |  0.25 |  0.452  |    0.733   |         62% |
> > > |          | MRR    |  0.318 | 0.116 |  0.424  |    0.711   |         68% |
> > > | A-V (L1) | NDCG   |  0.459 | 0.389 |   0.63  |    0.671   |          7% |
> > > |          | MRR    |  0.364 | 0.291 |  0.425  |    0.698   |         64% |
> > > | V-A (L1) | NDCG   |  0.283 | 0.235 |  0.397  |    0.584   |         47% |
> > > |          | MRR    |  0.133 | 0.088 |  0.251  |    0.586   |        133% |
> > > | P-A (L2) | NDCG   |  0.229 | o.o.m |  0.229  |    0.282   |         23% |
> > > |          | MRR    |  0.121 | o.o.m |  0.098  |    0.225   |        130% |
> > > | A-P (L2) | NDCG   |  0.197 | o.o.m |  0.202  |    0.287   |         42% |
> > > |          | MRR    |  0.095 | o.o.m |   0.08  |    0.242   |        203% |
> > > | A-V (L2) | NDCG   |  0.347 | o.o.m |  0.384  |    0.402   |          5% |
> > > |          | MRR    |  0.310 | o.o.m |  0.226  |    0.399   |         77% |
> > > | V-A (L2) | NDCG   |  0.235 | o.o.m |  0.222  |    0.252   |         14% |
> > > |          | MRR    |  0.13  | o.o.m |  0.087  |    0.166   |         91% |
> > >
> > > As shown in the table, our proposed KTN outperforms both baselines by up to 203%  higher in MRR across 8 different transfer learning tasks on the OAG-computer science graph. Especially, ALDA, which needs to compute a (k x k) confusion matrix for k classes, shows out-of-memory errors on L2 tasks where the number of classes is 17,729. We will add all those results and analyses to the final manuscript.
> > >
> > > [1] Chen et al., Adversarial-Learned Loss for Domain Adaptation, AAAI 2020
> > > [2] Wei et al., ToAlign: Task-oriented Alignment for Unsupervised Domain Adaptation, Neurips 2021
> > >
> > > **Q5. The experimental settings for Section 6.3 need more details.**
> > >
> > > First of all, we add a new Section 3.3 to define our new problem setting more rigorously as follows:
> > >
> > > **Problem Definition 1. Zero-shot cross-type transfer learning running on a HG:**
> > >     In a given heterogeneous graph \\(\mathcal{G} = (\mathcal{V}, \mathcal{E}, \mathcal{T}, \mathcal{R})\\) with node attributes \\(\mathcal{X} = \cup_{t\in \mathcal{T}}\mathcal{X}_t\\), assume node types \\(\mathbf{s}\\) and \\(\mathbf{t}\\) share a classification task \\(\{(i, y_i): i \in \mathcal{V}_s, \mathcal{V}_t\}\\).
> > >     During the training phase, using labels \\(\{(i, y_i): i \in \mathcal{V}_s\}\\) only for source-type nodes, we train an HGNN model \\(\textbf{f}: \textbf{f}(\mathcal{G}, \mathcal{X}) = h_i\\) to get node embeddings \\(h_i\\) for all nodes \\(i \in \mathcal{V}\\) and a classifier \\(\textbf{g}: \textbf{g}(h_i) = \hat{y}_i\\) to predict labels \\(\hat{y}_i\\) from the node embeddings \\(h_i\\).
> > >     During the test phase, our task is to predict labels \\(\{(j, y_j): j \in \mathcal{V}_t\}\\) of target-type nodes (where none of the labels of target-type nodes were used for training).
> > >
> > > Based on this problem definition, the experimental settings for Section 6.3 are as follows:
> > > During training, we are given 1) the heterogeneous graph structure information (i.e., adjacency matrices between same/different node types), 2) input node attribute matrices for all node types, and 3) labels on source-type nodes for the classification task. During the test phase, we predict labels on target-type nodes for the same classification task.
> > >
> > > We add these descriptions at the beginning of Section 6.3. For your information, details of the datasets we used in Section 6.3 are described in Appendix A.8; Details of domain adaptation baselines and HGNN models can be found in Appendix A.9 and A.10; Details of the model architecture and optimization are described in A.11 Experimental Settings.

---

> > > > ### Comment · Reviewer_deGV · 2022-08-08
> > > > **Thanks for the response.**
> > > >
> > > > Thank the authors for their response. I am satisfied with the response and the revised version. I would still recommend a positive score.

---

> > > > > ### Author Response · Authors · 2022-08-09
> > > > > **Thanks for the reviews and comments.**
> > > > >
> > > > > Thank you for your thoughtful reviews and comments.
> > > > > They helped a lot to improve the quality of our paper.

---

### Official Review · Reviewer_sHGC · 2022-07-15

**Rating:** 6
**Confidence:** 4
**Soundness:** 3 good
**Presentation:** 3 good
**Contribution:** 3 good

**Summary:**

This paper proposes a novel domain adaptation method on heterogeneous graphs by reconstructing the source node embeddings through target node embeddings. Theoretical proofs verify the correctness of proposed method. Experiments show that the proposed method achieves significant performance promotion compared with bare HGNN model.

**Questions:**

1. Authors should more clearly state how the gradients are computed in Figure 2 (b)-(c), i.e. they are gradients under loss $\mathcal{L}_s$.

**Limitations:**

yes

**Strengths And Weaknesses:**

Strengths：

- This paper is good structured and quite informative. Nice writing.
- Figure 1 gives a simple and clear example, demonstrating how the HGNN is trained with respect to different node types. The combination of Figure 1 and Figure 2 justifies the motivation of this paper.
- Theoretical results in Section 4.3 is intuitive, easy to understand. Theorem 1 induces an elegant algorithm in Section 5.
- Extensive experiments validate the effectiveness (Table 1 & 2) and generality (Table 3) of the proposed method.Weaknesses:

Weaknesses：
- The methods compared in Table 1 seem to be out-dated. Some recents works could be considered, i.e. Adversarial-Learned Loss for Domain Adaptation (AAAI2020).

---

> ### Author Response · Authors · 2022-08-02
> **Response to R1**
>
> Thank you for the valuable suggestions and references. We hereby carefully address your concerns as follows:
>
> **Q1. Comparison with more recent domain adaptation methods. E.g., AAAI 2020, adversarial-learned loss for domain adaptation**
>
> We run two more recent domain adaptation baselines, ALDA [1] (AAAI 2020) and ToAlign [2] (Neurips 2021), on the zero-shot transfer learning tasks on the OAG graph (the same experiments we described in Table 1 in Section 6.3).
>
> ALDA [1] combines domain-adversarial learning and self-training to reduce the gap and align the feature distributions. It learns a confusion matrix through an adversarial manner to correct the pseudo-label of the unlabeled target sample. ToAlign [2] makes the domain alignment proactively serve classification by aligning target features with source task-discriminative features, which are obtained under the guidance of meta-knowledge induced from the classification task. Both ALDA and To Align assume the source and target domains share the same feature extractors (as all other DA methods assume). This assumption results in poor DA performance on HGNNs with a unique feature extractor for each domain.
>
> |   Task   | Metric | Source |  ALDA | ToAlign | KTN (ours) | Improvement |
> |:--------:|:------:|:------:|:-----:|:-------:|:----------:|:-----------:|
> | P-A (L1) | NDCG   |  0.399 | 0.235 |  0.472  |    0.623   |         32% |
> |          | MRR    |  0.297 | 0.096 |  0.451  |    0.629   |         39% |
> | A-P (L1) | NDCG   |  0.401 |  0.25 |  0.452  |    0.733   |         62% |
> |          | MRR    |  0.318 | 0.116 |  0.424  |    0.711   |         68% |
> | A-V (L1) | NDCG   |  0.459 | 0.389 |   0.63  |    0.671   |          7% |
> |          | MRR    |  0.364 | 0.291 |  0.425  |    0.698   |         64% |
> | V-A (L1) | NDCG   |  0.283 | 0.235 |  0.397  |    0.584   |         47% |
> |          | MRR    |  0.133 | 0.088 |  0.251  |    0.586   |        133% |
> | P-A (L2) | NDCG   |  0.229 | o.o.m |  0.229  |    0.282   |         23% |
> |          | MRR    |  0.121 | o.o.m |  0.098  |    0.225   |        130% |
> | A-P (L2) | NDCG   |  0.197 | o.o.m |  0.202  |    0.287   |         42% |
> |          | MRR    |  0.095 | o.o.m |   0.08  |    0.242   |        203% |
> | A-V (L2) | NDCG   |  0.347 | o.o.m |  0.384  |    0.402   |          5% |
> |          | MRR    |  0.310 | o.o.m |  0.226  |    0.399   |         77% |
> | V-A (L2) | NDCG   |  0.235 | o.o.m |  0.222  |    0.252   |         14% |
> |          | MRR    |  0.13  | o.o.m |  0.087  |    0.166   |         91% |
>
> As shown in the table above, our proposed KTN outperforms both baselines by up to 203%  higher in MRR across 8 different transfer learning tasks on the OAG-computer science graph. Especially, ALDA, which needs to compute a \\((k \times k)\\) confusion matrix for \\( k\\) classes, shows out-of-memory errors on L2 tasks where the number of classes is 17,729. We will add all those results and analyses to the final manuscript.
>
> [1] Chen et al., Adversarial-Learned Loss for Domain Adaptation, AAAI 2020
> [2] Wei et al., ToAlign: Task-oriented Alignment for Unsupervised Domain Adaptation, Neurips 2021
>
> **Q2. How are the gradients computed in Figures 2 (b)-(c)?**
>
> In Figures 2 (b-c), we show that feature extractors of different node types are using different sets of parameters of the same HGNN model. Thus computation (gradient) paths become unique for each node type. Figure 2(b) shows that when source labels are given, how the gradients of a loss \\(\mathcal{L}_s\\) (computed using output embeddings of source nodes and source labels) are propagated through the HGNN. On the contrary, Figure 2(c) presents how gradients of a target loss \\(\mathcal{L}_t\\) (computed using target embeddings and target labels) are propagated through the HGNN model.
>
> Our zero-shot transfer learning setting has no target labels during the training time. In other words, during the training time, we pass gradients through the red gradient path using source labels, as shown in Figure 2(b). During the test stage, we're using the parameters related to target node types (to predict target labels), which don't get a chance to be updated via the blue gradient path in Figure 2(c) during training time (as we don't have target labels in training time). As the different computation (gradient) paths are used between the training and testing phases, an HGNN model pretrained on the source domain shows a poor performance on the target domain.

---

### Meta-Review · Area_Chair_Wo12 · 2022-08-25

**Recommendation:** Accept
**Confidence:** Certain

**Metareview:**

This paper proposes a transfer learning technique in heterogenous graphs, which can contain different types of nodes and edges. It identifies a limitation of existing work on graph neural networks for heterogenous graphs: they tend to implicitly learn separate representation encoders for each type of node. In other words, even though the same network is applied to all nodes, it tends to use separate activation paths to compute the representations. The proposed Knowledge Transfer Networks (KTNs) address this limitation with a novel architecture that explicitly learns a mapping between different domains (in this case, node types). Experiments on benchmark heterogeneous graphs show that KTNs lead to large improvements in classification accuracy on new node types.

The reviewers all liked the contributions of the paper, identifying the problem as important, the KTN architecture as novel, and the experimental results convincing. During the review process, the authors have already updated the paper several times in response to the reviewers feedback. In addition, they are encouraged to include the additional results with ALDA in the final version.

One remaining area for improvement is precisely scoping in the introduction and informal problem statements what exactly is meant by zero-shot learning in this context. As stated in the limitation statement, KTNs require that the task share the same label space. The authors have clarified this in section 3.3, but that is still too late for such a key definition.

**Award:**

No

---

### Decision · Program_Chairs · 2022-09-14

Accept